# Effect of very large body mass loss on energetics, mechanics and efficiency of walking in adults with obesity: mass-driven *versus* behavioural adaptations

Davide Malatesta[1] 🆔, Julien Favre[2], Baptiste Ulrich[2], Didier Hans[3], Michel Suter[4], Lucie Favre[5] and Aitor Fernández Menéndez[1]

[1] *Institute of Sport Sciences of the University of Lausanne (ISSUL), University of Lausanne, Lausanne, Switzerland*
[2] *Swiss BioMotion Lab, Department of Musculoskeletal Medicine, Lausanne University Hospital and University of Lausanne (CHUV-UNIL), Lausanne, Switzerland*
[3] *Center for Bone Diseases, Lausanne University Hospital, Lausanne, Switzerland*
[4] *CHUV, Service de chirurgie viscérale, Bâtiment hospitalier du CHUV, Lausanne, Switzerland*
[5] *Consultation de prévention et traitement de l'obésité, Lausanne University Hospital (CHUV), Lausanne, Switzerland*

Edited by: Michael Hogan & Audrey Bergouignan

Linked articles: This article is highlighted in a Journal Club article by Luciano *et al*. To read this article, visit https://doi.org/10.1113/JP282385.

The peer review history is available in the Supporting Information section of this article (https://doi.org/10.1113/JP281710#support-information-section).

**Davide Malatesta** is a senior lecturer at the Institute of Sport Sciences of the University of Lausanne. He received a BSc, MSc and PhD in Human Movement Sciences. He is interested in the energetics and biomechanics of locomotion in healthy and pathological conditions. His current research focuses on the effect of obesity and ageing on walking economy and mechanics.

**Abstract** Understanding the mechanisms involved in the higher energy cost of walking ($NC_w$: the energy expenditure above resting per unit distance) in adults with obesity is pivotal to optimizing the use of walking in weight management programmes. Therefore, this study aimed to investigate the mechanics, energetics and mechanical efficiency of walking after a large body mass loss induced by bariatric surgery in individuals with obesity. Nine adults ($39.5 \pm 8.6$ year; BMI: $42.7 \pm 4.6$ kg m$^{-2}$) walked at five fixed speeds before (baseline) and after the bariatric surgery (post 1 and post 2). Gas exchanges were measured to obtain $NC_w$. A motion analysis system and instrumented treadmill were combined to assess total mechanical work ($W_{tot}$). Mechanical efficiency ($W_{tot} NC_w^{-1}$) was also calculated. Participants lost $25.7 \pm 3.4\%$ of their body mass at post 1 (6.6 months; $P < 0.001$) and $6.1 \pm 4.9\%$ more at post 2 (12 months; $P = 0.014$). Mass-normalized $NC_w$ was similar between baseline and post 1 and decreased at post 2 compared to that at baseline ($-6.2 \pm 2.7\%$) and post 1 ($-8.1 \pm 1.9\%$; $P \leq 0.007$). No difference was found in mass-normalized $W_{tot}$ during follow-up ($P = 0.36$). Mechanical efficiency was similar at post 1 and post 2 when compared to that at baseline ($P \geq 0.19$), but it was higher ($+14.1 \pm 4.6\%$) at post 2 than at post 1 ($P = 0.013$). These findings showed that after a very large body mass loss, individuals with obesity may reorganize their walking pattern into a gait more similar to that of lean adults, thus decreasing their $NC_w$ by making their muscles work more efficiently.

(Received 31 March 2021; accepted after revision 26 August 2021; first published online 10 September 2021)
**Corresponding author** D. Malatesta: University of Lausanne (UNIL), Institute of Sport Sciences of University of Lausanne, Bâtiment Synathlon, 1015 Lausanne, Switzerland. Email: Davide.Malatesta@unil.ch

**Abstract figure legend** The very large body mass loss induced by bariatric surgery significantly decreased the mass-normalized energy cost of walking only after 1 year with similar relative total mechanical work and, thus, with an increased mechanical efficiency. However, the mechanical and energetic adaptations to body mass loss were time dissociated during the 1-yr follow-up. The mechanical adaptations (1) emerged quickly and was a mainly mass-driven adaptation, whereas the energetic adaptations (2) appeared only after a period of adaptation to the new gait pattern as a more adaptive and behavioural change.

### Key points

- A higher net (above resting) energy cost of walking (lower gait economy) is observed in adults with obesity compared to lean individuals.
- Understanding the mechanisms (i.e. mass driven, gait pattern and behavioural changes) involved in this extra cost of walking in adults with obesity is pivotal to optimizing the use of walking to promote daily physical activity and improve health in these individuals.
- We found that very large weight loss induced by bariatric surgery significantly decreased the energy cost of walking per kg of body mass after 1 year with similar total mechanical work per kg of body mass, resulting in an increased mechanical efficiency of walking.
- Individuals with obesity may reorganize their walking pattern into a gait more similar to that of adults of normal body mass, thus decreasing their energy cost of walking by making their muscles work more efficiently.

## Introduction

Obesity has been recognized as a significant public health issue worldwide, with a prevalence that has been continually increasing over the past decades, leading to a variety of chronic diseases and increasing health care costs (NCD Risk Factor Collaboration, 2016). The causes of obesity are multifactorial, but the energy imbalance (i.e. lower daily energy expenditure compared to daily energy intake) seems to be the main driver leading to weight gain (Yoo, 2018). Although a reduction in physical activity level (Guthold *et al.* 2018) may be involved in a decreased daily energy expenditure, the role of physical activity in energy balance (Blundell *et al.* 2015) and weight loss remains controversial (Pontzer *et al.* 2016). However, independently of weight loss, physical activity is crucially important for improving overall health and fitness in the prevention and treatment of obesity (Luke & Cooper, 2013).

Walking is the most common modality to promote daily physical activity, reduce sedentary time and improve health (Murtagh *et al.* 2015). However, the higher net energy cost of walking ($NC_w$: the energy expended above the resting energy expenditure per unit distance) in individuals with obesity than in their lean counterparts may lead to motor impairments and fatigue and contribute to an increase in physical inactivity and sedentary time in daily life in the former (Levine *et al.* 2005; Levine *et al.* 2008). This may thus reduce the efficacy of walking in weight management programmes. Moreover, the absolute $NC_w$ ($J\ m^{-1}$) but also the relative $NC_w$ (i.e. normalized by body mass; $J\ kg^{-1}\ m^{-1}$) are higher in individuals with obesity than in lean adults (Browning *et al.* 2006; Browning & Kram, 2007; Fernandez Menendez *et al.* 2020), suggesting that body mass is the main but not the only factor affecting the lower walking economy in this population. Therefore, understanding the mechanisms (i.e. mass driven, gait pattern and behavioural changes) involved in this extra cost of walking in adults with obesity is pivotal to improving gait economy and optimizing the use of walking to promote daily physical activity and improve health in these individuals.

Recently, Fernandez Menendez *et al.* (2020) showed that the relative $NC_w$ was 19% higher in adults with class III obesity than in their lean counterparts and was associated with a lower amount of mass-normalized external mechanical work (i.e. work performed to lift and accelerate the centre of mass (CM) relative to the surroundings, $W_{ext}$), higher pendular recovery (i.e. gait energy saving mechanism due to the exchange between potential and kinetic energy of CM that minimizes $W_{ext}$) but similar mass-normalized internal (i.e. work required to move the limbs with respect to CM, $W_{int}$) and total (i.e. $W_{tot} = W_{ext} + W_{int}$) mechanical work. As a consequence, the mechanical efficiency ($W_{tot}\ NC_w^{-1}$) was reduced in individuals with obesity due to their higher relative $NC_w$, likely related to muscle level differences (e.g. more muscle fibre work or force and/or poorer muscle efficiency in individuals with obesity than in lean adults). This may be associated with the more erect gait pattern (i.e. reduced hip and knee flexion and increased ankle plantar flexion) (DeVita & Hortobagyi, 2003; Fernandez Menendez *et al.* 2018), which requires larger muscle activation and makes muscles operate in disadvantageous lengths and/or velocities, thereby inducing poorer muscle efficiency (Massaad *et al.* 2007) in individuals with obesity than in lean individuals.

This more erect gait pattern in adults with class II and III obesity is reversible and becomes more dynamic due to increased hip range of motion, knee flexion and ankle function (mechanical plasticity) after a very large body mass loss (−34%) induced by bariatric surgery with a body mass limiting gait change threshold of 30 kg (~25% of the initial body mass) (Hortobagyi *et al.* 2011).

This mechanical plasticity also characterizes the improved pendular transduction between the mechanical energies during walking in response to obesity (Malatesta *et al.* 2013; Fernandez Menendez *et al.* 2019*b*, 2020) or load in African women carrying 20% of their body mass on their heads (Heglund *et al.* 1995). However, this more skilful pendular mechanism is lost in unloading walking conditions in African women (Heglund *et al.* 1995), showing that this mechanism is also reversible. Therefore, a very large body mass loss obtained with bariatric surgery may modify the gait pattern, making it more dynamic with higher muscle efficiency and less pendulum movement. These changes may be involved in a decreased absolute and relative $NC_w$ obtained with body mass loss, as previously shown in healthy adolescent individuals with obesity after only 6% body mass loss after an obesity management programme (Peyrot *et al.* 2010). This reduction in $NC_w$ was mainly associated with decreased body mass but also with changes in the biomechanical parameters of walking (i.e. less lower limb muscle work required to rise the CM with mass-normalized $W_{ext}$ unchanged after intervention). The authors hypothesized that the relation between the changes in absolute $NC_w$ and the changes in the biomechanical parameters might be explained by an increase in the efficiency of muscle mechanical work with body mass loss, as previously shown in cycling (Rosenbaum *et al.* 2003). However, this enhancement of muscle efficiency during walking after a very large body mass loss in adults has not yet been investigated.

The purpose of this study was to investigate the mechanics, energetics and mechanical efficiency of walking after a very large body mass loss induced by bariatric surgery in adults with obesity (1 year follow-up). We tested these changes at five fixed and equally spaced walking speeds at three time points: before, midway (−25% of initial body mass: ~6 months) and after ~1 year of surgery. It was hypothesized that bariatric surgery may decrease the mass-normalized $NC_w$ as a consequence of a reduction in the total body mass as well as an increase in mechanical efficiency, which would allow the muscles to provide similar mass-normalized total mechanical work in favourable conditions. Moreover, compared to 1 year after surgery, these mass-driven and adaptive gait changes would be greater at the midway time point because the limiting gait changes threshold and greater relative body mass loss were reached at the midway time point.

## Methods

### Ethical approval

The study was approved by the local ethics committee (CER-VD 2016-01715). The study conformed to the standards set by the *Declaration of Helsinki*, except for

**Table 1. Changes in anthropometrics and standing metabolic rate after bariatric surgery**

| | Baseline ($n = 9$) | Post 1 ($n = 9$) | Post 2 ($n = 9$) |
|---|---|---|---|
| BMI (kg m$^{-2}$) | 42.7 ± 4.6 | 31.7 ± 4.4* | 29.5 ± 6.5*,† |
| Body mass (kg) | 114.7 ± 10.5 | 85.3 ± 10.1* | 78.6 ± 14.5*,† |
| Lean body mass (kg) | 52.4 ± 5.4 | 45.6 ± 5.5* | 45.7 ± 5.6* |
| Fat body mass (kg) | 58.1 ± 11.5 | 36.1 ± 10.6* | 29.9 ± 15.5*,† |
| Fat body mass (%body mass) | 50.3 ± 6.3 | 41.8 ± 8.1* | 35.9 ± 12.2*,† |
| Android/gynoid | 1.24 ± 0.2 | 1.15 ± 0.1 | 1.03 ± 0.1* |
| VAT (cm$^3$) | 2.58 ± 1.2 | 1.11 ± 0.4* | 0.82 ± 0.5* |
| Lower limb mass (kg) | 39.5 ± 6.9 | 28.9 ± 4.6* | 27.9 ± 6.1* |
| Lean lower limb mass (kg) | 19.8 ± 1.9 | 16.0 ± 2.1* | 16.2 ± 1.5* |
| Fat lower limb mass (kg) | 18.8 ± 6.8 | 11.9 ± 4.2* | 10.8 ± 5.8* |
| Upper limb mass (kg) | 12.1 ± 1.2 | 8.8 ± 1.6* | 8.5 ± 1.8* |
| Lean upper limb mass (kg) | 5.9 ± 0.9 | 4.7 ± 1.1* | 4.8 ± 0.9* |
| Fat upper limb mass (kg) | 5.9 ± 1.6 | 3.7 ± 1.2* | 3.4 ± 1.8* |
| Trunk mass (kg) | 59.0 ± 6.4 | 44.6 ± 9.4* | 41.2 ± 12.3* |
| Lean trunk mass (kg) | 26.7 ± 3.0 | 24.8 ± 3.4* | 24.7 ± 3.5* |
| Fat trunk mass (kg) | 33.4 ± 5.1 | 20.4 ± 7.7* | 15.7 ± 8.4*,† |
| SMR (W) | 131.4 ± 15.5 | 98.2 ± 19.7* | 107.8 ± 15.4* |
| Adjusted SMR‡ (W) | 117.1 ± 23.5 | 103.1 ± 16.5 | 117.1 ± 18.2 |
| SMR (W kg$^{-1}$) | 1.15 ± 0.11 | 1.16 ± 0.21 | 1.40 ± 0.22*,† |
| Adjusted SMR§ (W) | 125.6 ± 18.3 | 101.3 ± 16.5* | 108.3 ± 16.2 |
| SMR (W kg$^{-1}$ lean body mass) | 2.52 ± 0.33 | 2.15 ± 0.34* | 2.33 ± 0.34 |

Values are the mean ± SD ($n = 9$, except for body composition assessment at post 2: $n = 8$). Baseline, prebariatric surgery; post 1, −25% of initial body mass (∼6.6 months average) after bariatric surgery; post 2, 12 months after bariatric surgery.
*Significant difference from baseline ($P < 0.05$).
†Significant difference from post 1 ($P < 0.05$).
‡Adjusted values with body mass as covariate.
§Adjusted values with lean body mass as covariate. BMI, body mass index; SMR, standing metabolic rate; VAT, visceral adipose tissue.

registration in a database. All the subjects provided written informed consent.

## Participants

Nine sedentary adults with obesity (39.5 ± 8.6 year; ≤2 h of physical activity per week over the past year) were recruited to participate in this study (Table 1). All participants were healthy and free of musculoskeletal injuries and cardiovascular and respiratory diseases that could affect their gait pattern. Inclusion criteria included obesity (BMI > 35 kg m$^{-2}$), age (range 18–60 years), mobility and pain-free status throughout the study. Based on a medical exam, the exclusion criteria were age >60 year, BMI >60 kg m$^{-2}$, neurological disorders, orthopaedic injury, cardiovascular diseases, history of falls and medications that provoke dizziness.

## Experimental design

Participants reported to the laboratory on three occasions. The first session was scheduled before the bariatric surgery performed at the Lausanne University Hospital (CHUV) (baseline), while the second and third sessions were scheduled when each participant decreased their body mass by ∼25% (average body mass loss expected ∼6 months after the surgery: the body mass loss threshold for gait changes (Hortobagyi *et al.* 2011)) (post 1) and 1 year after the bariatric surgery (post 2). During each session, body mass and composition through dual-energy X-ray absorptiometry (DXA) as well as metabolic and mechanical data during 5 min treadmill-walking bouts at five different and equally spaced speeds (0.56, 0.83, 1.11, 1.39, and 1.67 m s$^{-1}$), separated by 5 min resting periods were obtained. Participants were asked to complete the walking trials without using handrail support. The order of the speeds was determined randomly and maintained at each point of the follow-up for each participant. To standardize the pre-exercise conditions, participants were asked to avoid strenuous exercise the day before each experimental trial, and they reported to the laboratory after a minimum 3 h fast period and at a similar time of day to avoid circadian variance.

### Assessments

**Body composition and anthropometric characteristics.** DXA (GE Healthcare Lunar, Madison, WI, USA) was used to assess total and regional body mass, lean and fat mass as well as the height and width of each anthropometric segment (hand, forearm, upper arm, foot, shank, thigh, head and trunk). These measurements were used to obtain a personalized mathematical model for each participant that represents the individual's body segments as simple geometrical solids to determine the centre of mass (CM) and inertial properties of each segment as previously described (Hanavan, 1964) and used in the same population (Fernandez Menendez *et al.* 2020).

**Physical activity level.** Each participant completed a self-reported measurement of a habitual physical activity questionnaire assessing physical activity at work and physical activity during leisure, excluding sports and sports during leisure time over 6 months before the baseline, between the baseline and post 1 and post 1 and post 2. Different scores were used to quantify work, leisure and sport activities, altogether resulting in a total physical activity score (Baecke *et al.* 1982).

**Preferred walking speed.** All the participants performed a 10 min walking familiarization on an instrumented single-belt treadmill (T150-FMT-MED, Arsalis, Louvain-la-Neuve, Belgium) at the experimental walking speeds (Wall & Charteris, 1981). Then, the preferred walking speed (PWS) was determined according to a protocol previously described (Martin *et al.* 1992) and used with individuals with obesity (Fernandez Menendez *et al.* 2019*a*). Briefly, participants started to walk on the treadmill at the lowest experimental speed ($0.56$ m s$^{-1}$) without receiving any feedback regarding their speed. The speed was gradually increased until the participant subjectively identified the PWS and maintained this for 1 min, slightly modified according to the participant's instructions. This procedure was repeated starting from the highest experimental speed ($1.67$ m s$^{-1}$) or from the previously assessed PWS + $0.42$ m s$^{-1}$ (when PWS >$1.25$ m s$^{-1}$) and then the speed was gradually reduced to the individual subjective PWS. The average of the two speeds selected by each participant (i.e. increasing and decreasing speed trials) was considered the final PWS.

**Energy cost of walking.** Expired gases (oxygen uptake ($\dot{V}_{O_2}$) and $CO_2$ output ($\dot{V}_{CO_2}$)) were collected (Quark CPET, Cosmed, Albano Laziale, Italy) breath-by-breath in the standing position for 5 min or more, until the experimenters visually determined when the stability of $\dot{V}_{O_2}$ and $\dot{V}_{CO_2}$ was reached for each participant. This stability was further visually checked by two blinded and independent investigators who selected the last minute or 1-min time stable window during the data analysis to average $\dot{V}_{O_2}$ and $\dot{V}_{CO_2}$ values and assess the standing metabolic rate during resting (SMR). Breath-by-breath $\dot{V}_{O_2}$ and $\dot{V}_{CO_2}$ were measured during each walking speed with a respiratory exchange ratio (RER) of less than 1 for all participants and conditions. Volume and gas calibrations were performed before each trial. Breath-by-breath $\dot{V}_{O_2}$ data were initially examined to exclude errant breaths due to coughing or swallowing, and values that were more than 3 standard deviations (SDs) from the local mean were discarded. $\dot{V}_{O_2}$ values (ml $O_2$ kg$^{-1}$ min$^{-1}$) from the last minute (i.e. steady state) were averaged. With the average RER over the same last minute, the energy equivalent of 1 litre of $O_2$ was determined [J (l $O_2$)$^{-1}$] (Astrand, 1986) and then multiplied by the average $\dot{V}_{O_2}$ (l $O_2$ kg$^{-1}$ s$^{-1}$) at steady state during the last minute of walking for each speed to calculate the gross metabolic rate in W kg$^{-1}$. The same procedure was applied to convert the average $\dot{V}_{O_2}$ during standing in W kg$^{-1}$. The net metabolic rate (W kg$^{-1}$) was calculated by subtracting the SMR from the gross metabolic rate and then divided by the corresponding walking speed (m s$^{-1}$) to determine NC$_w$. This latter was expressed in absolute (J m$^{-1}$) and relative (i.e. normalized by the body mass (J kg$^{-1}$ m$^{-1}$) and lean body mass (J kg lean mass$^{-1}$ m$^{-1}$)) values throughout this article. SMR is commonly used to calculate NC$_w$ (Martin *et al.* 1992; Donelan *et al.* 2002; Browning *et al.* 2006; Browning & Kram, 2007; Massaad *et al.* 2007; Fernandez Menendez *et al.* 2020) and, compared with basal metabolic rate, it better defines the metabolic cost associated with muscle contractions to maintain balance and support body weight while walking (Malatesta *et al.* 2003), which have to be considered for accurately assessing NC$_w$ in individuals with normal weight (Malatesta *et al.* 2003) and with obesity (Peyrot *et al.* 2012).

**External mechanical work, spatiotemporal parameters and recovery.** An instrumented treadmill was used to assess $W_{ext}$ according to the methodology described in previous studies (Fernandez Menendez *et al.* 2018, 2019*b*, 2020). Twenty steps from the last 30 s of each walking trial were selected to obtain the vertical ($F_v$), forward ($F_f$), and lateral ($F_l$) ground reaction forces (1000 Hz sampling rate). The beginning and end of each step were defined as the instant when $F_f$ was equal to zero (Fernandez Menendez *et al.* 2020). Step length and duration as well as single and double support durations were then assessed. The 3D accelerations of CM were computed from ground reaction forces and the mass of the subjects. The mathematical integration of the 3D accelerations gave the velocity changes of the CM in the three directions ($V_v$, $V_f$ and $V_l$). From the instantaneous CM velocities and body mass ($m$), we computed the instantaneous vertical,

forward and lateral kinetic energies of the CM ($E_{kv}$, $E_{kf}$ and $E_{kl}$, respectively).

$$E_k = E_{kf} + E_{kv} + E_{kl} = 0.5m \left( V_f^2 + V_v^2 + V_l^2 \right) \quad (1)$$

A second mathematical integration of $V_v$ was performed to determine the vertical displacement of the CM ($h$), and the instantaneous gravitational potential energy ($E_p$) was computed from $h$, $m$ and gravity ($g = 9.81$ m s$^{-2}$).

$$E_p = mgh \quad (2)$$

Total mechanical energy ($E_{tot}$) was calculated as the sum of the increments in $E_k$ and $E_p$.

$$E_{tot} = E_k + E_p = E_{kf} + E_{kv} + E_{kl} + E_p \quad (3)$$

The sum of positive increments of $E_{tot}$ was equal to the amount of $W_{ext}$ performed per step. The fraction of mechanical energy recovered (Recovery) due to the pendulum mechanism (i.e. pendular transduction of potential into kinetic energy and vice versa) was obtained as follows:

$$\text{Recovery (\%)} = \frac{W_k + W_p + W_k^- + W_p^- - W_{ext} - W_{ext}^-}{W_k + W_p + W_k^- + W_p^-}$$
$$\times 100 \quad (4)$$

where $W_k$, $W_p$ and $W_{ext}$ represent the sum of the increments in the $E_k$, $E_p$ and $E_{tot}$ curves, respectively, and $W_k^-$, $W_p^-$ and $W_{ext}^-$ represent the sum of the decrements in the $E_k$, $E_p$ and $E_{tot}$ curves, respectively.

**Internal mechanical work.** The specific methodology used to assess the internal mechanical work in individuals with obesity was previously described (Fernandez Menendez *et al.* 2020). Briefly, a motion capture system based on optical technology (100 Hz sampling rate) with a set of eight infrared cameras (Smart-DX, BTS Bioengineering Corp., Garbagnate Milanese, Italy) was synchronized with the instrumented treadmill and used to collect kinematic and kinetic data for each step selected. Reflective markers were placed on both sides of the body over the following anatomical landmarks identified via DXA (Fernandez Menendez *et al.* 2020): seventh cervical vertebra, right scapular inferior angle, acromion, humerus, humeral lateral epicondyle, ulnar styloid, posterior and superior iliac spines, greater trochanter, medial and lateral epicondyles of the femur, medial and lateral malleoli, calcaneus and second metatarsal. Clusters of four non-collinear markers were positioned on the thigh, shank and sacrum. The coordinates and trajectories of all the markers during the walking trials were recorded and computed to obtain the linear velocity of the CM of the $i$th segment ($V_i$) and its angular velocity ($w_i$). The kinetic energy ($E_{kint}$) due to the movements of the segments relative to the body CM was calculated as the sum of the

translational and rotational kinetic energy for each step (Eq. 5):

$$E_{Kint} = \frac{1}{2} m_i V_{ri}^2 + \frac{1}{2} m_i k_i^2 w_i^2 \quad (5)$$

where $m_i$ and $k_i$ are the mass and radius of gyration of the $i$th segment obtained from the DXA; and $V_{ri}$ is the translational velocity of the CM of the $i$th segment relative to the body CM calculated by subtracting the absolute velocity of the body CM (obtained from the ground reaction forces) from $V_i$. To minimize errors due to noise in the signals, the $E_{kint}$ signal was low pass filtered with a fourth-order zero-lag Butterworth filter and a cut-off frequency of 7 Hz. Points identified as outliers were corrected using a spline interpolation method. To account for the energy transfer between segments, the $E_{kint}$ curves of the segments of the same limb were summed. $W_{int}$ for each limb was calculated as the sum of positive increments of the respective $E_{kint}$ curve, and the total $W_{int}$ was computed as the sum of the $W_{int}$ of the four limbs and that of the head-trunk segment (i.e. no energy transfer among limbs).

**Total mechanical work.** The total positive mechanical work performed per distance travelled ($W_{tot}$) was assessed as the sum of $W_{ext}$ and $W_{int}$, assuming no transfer of energy between the two types of energy (Willems *et al.* 1995). Throughout this article, all the mechanical work values are expressed as both absolute (J m$^{-1}$) and relative (i.e. normalized by body mass, J kg$^{-1}$ m$^{-1}$) values.

**Mechanical efficiency.** Mechanical efficiency was computed as the ratio of $W_{tot}$ to $NC_w$.

**Joint kinematic parameters.** The range of motion (ROM) of hip, knee and ankle joints in the sagittal plane was assessed using the motion capture system. For the hip joint, the neutral position (0°) was reached when the seventh cervical vertebra, the great trochanter and lateral condyle marker projections on the sagittal plane were completely aligned. The values are defined as positive for flexion and negative for extension. The neutral position of the knee (0°) corresponded to the alignment of the great trochanter, lateral condyle and malleolar projections on the sagittal plane. The values are defined as positive for flexion and negative for extension. The neutral position (0°) of the ankle joint was reached when the projection of the segments defined by the metatarsal-malleolus and the lateral condylar-malleolar projection markers formed 90° in the sagittal plane. Negative values represented ankle dorsiflexion, while positive values corresponded to plantar flexion movement. The average joint angular position of the hip, knee and ankle was assessed in the stance phase as well as in the swing phase only for the hip. Knee flexion

at heel strike and maximal knee flexion in the early stance (~15% of the cycle duration) and their difference (delta knee ROM) were also determined.

## Statistical analysis

Statistical analysis was performed using SPSS software version 25 (IBM Corp., Armonk, NY, USA).

Anthropometrics, physical activity level, SMR and PWS from baseline and at the two follow-ups were compared with a linear mixed model with participants set as a random effect. An updated linear mixed model with body mass as covariate and one with lean body mass as covariate were used for the absolute SMR. The mechanics and energetics of walking at five fixed speeds were evaluated with a linear mixed effects analysis of the relationships between conditions (walking speed (0.56, 0.83, 1.11, 1.39 and 1.67 m s$^{-1}$) and time (baseline *vs.* post 1 *vs.* post 2)). The fixed effects included walking speed and time, while the participants were set as a random effect. The normality of the residuals was tested using the Kolmogorov–Smirnov test. Because it is well accepted that speed influences metabolic and mechanical variables, the main effects of speed are not reported in this article. An updated linear mixed effects analysis of the relationships between conditions and time with body mass as a covariate and one with lean body mass were used for the absolute $NC_w$. To better understand the mechanisms related to $NC_w$, we performed Spearman's correlation analyses between the mechanical variables and $NC_w$. The level of significance was set to $P \leq 0.05$. All the values are reported as the mean ± SD.

## Results

### Participant characteristics

Following bariatric surgery, body mass significantly decreased at post 1 (i.e. 6.6 months averaged; $-25.7 \pm 3.4\%$; $P < 0.0001$) and post 2 (i.e. 12 months; $-31.9 \pm 8.1\%$; $P < 0.0001$) but also between the two follow-ups ($-8.5 \pm 6.8\%$; $P = 0.014$; Tables 1 and S1). Only one participant decreased his body mass less than the 25% expected at post 1 obtaining $-20\%$ at this time point.

The reduction in body mass was obtained with a significant decrease in body fat mass at post 1 ($-38.7 \pm 7.3\%$; $P < 0.0001$) and at post 2 ($-50.5 \pm 16.7\%$; $P < 0.0001$) and between the two follow-ups ($21.8 \pm 18.5\%$; $P = 0.015$) and with a significant reduction in lean body mass only at post 1 ($-13.1 \pm 3.4\%$; $P < 0.0001$) and post 2 ($-12.3 \pm 2.4\%$; $P < 0.0001$) with no significant difference between the two follow-ups ($P = 1$; Tables 1 and S1). Fat distribution changed

after bariatric surgery, with a significantly lower ratio of android/gynoid only between baseline and post 2 ($-15.7 \pm 11.8\%$; $P = 0.005$; Tables 1 and S1). Visceral adipose tissue was significantly lower between both follow-ups and the baseline ($P < 0.001$ for both) with no significant difference between post 1 and post 2 ($P = 0.92$; Tables 1 and S1). BMI significantly decreased between baseline and the two follow-ups ($P < 0.001$ for both) and between post 1 and post 2 ($P = 0.033$; Tables 1 and S1).

The mass, lean mass and fat mass of the lower and upper limbs as well as the mass and lean mass of the trunk significantly decreased at post 1 and post 2 compared to baseline ($P < 0.0001$ for all; Tables 1 and S1), with no significant differences between the two follow-ups ($P \geq 0.43$). Similarly, fat trunk mass significantly decreased at post 1 and post 2 after bariatric surgery ($P \leq 0.001$ for both) but also between post 1 and post 2 ($P = 0.017$; Tables 1 and S1).

### Physical activity level and preferred walking speed

There was no significant difference in physical activity score during follow-up (baseline: $7.7 \pm 1.5$, post 1: $7.8 \pm 1.3$ and post 2: $8.1 \pm 1.2$; $P = 0.52$; Table S1).

PWS was significantly faster at post 1 and post 2 than at baseline (baseline: $1.07 \pm 0.1$ m s$^{-1}$, post 1: $1.30 \pm 0.1$ m s$^{-1}$ and post 2: $1.32 \pm 0.1$ m s$^{-1}$; $P \leq 0.001$), with no significant difference between the two follow-ups ($P = 1$; Table S3).

### Energetics

The absolute SMR was significantly decreased at post 1 and post 2 compared with that at baseline ($P < 0.0001$ and $P = 0.007$, respectively), with no significant difference between the two follow-ups ($P = 0.50$; Tables 1 and S2). At post 2, the relative SMR per kg of body mass was significantly increased compared with that at baseline and post 1 ($P = 0.04$ and $P = 0.045$, respectively) with no significant difference between baseline and post 1 ($P = 1$; Table 1). The mixed linear model with body mass as covariate ($P = 0.042$) indicated that there was no significant difference in the absolute SMR during follow-up ($P = 0.13$; Table 1). The relative SMR per kg of lean body mass was significantly decreased only at post 1 compared with that at baseline ($P = 0.048$) with no significant difference between baseline and post 2 and between the two follow-ups ($P = 0.49$ and $P = 0.81$, respectively; Table 1). The mixed linear model with lean body mass as covariate ($P = 0.11$) showed that the absolute SMR was significantly decreased at post 1 compared to the baseline ($P = 0.02$) with no difference between baseline and post 2 ($P = 0.14$) and between the two follow-ups ($P = 0.95$; Table 1).

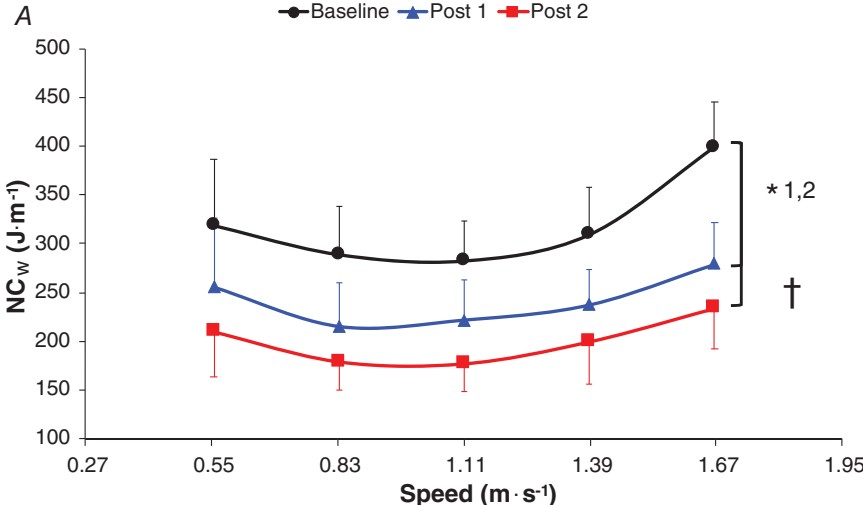

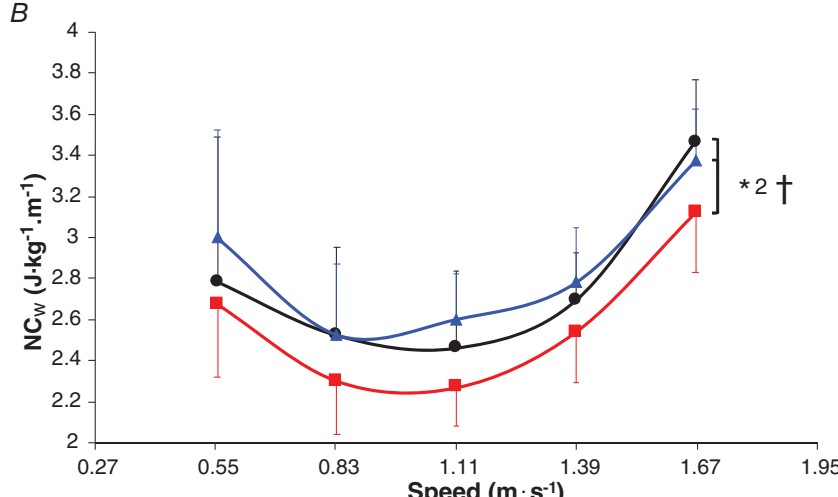

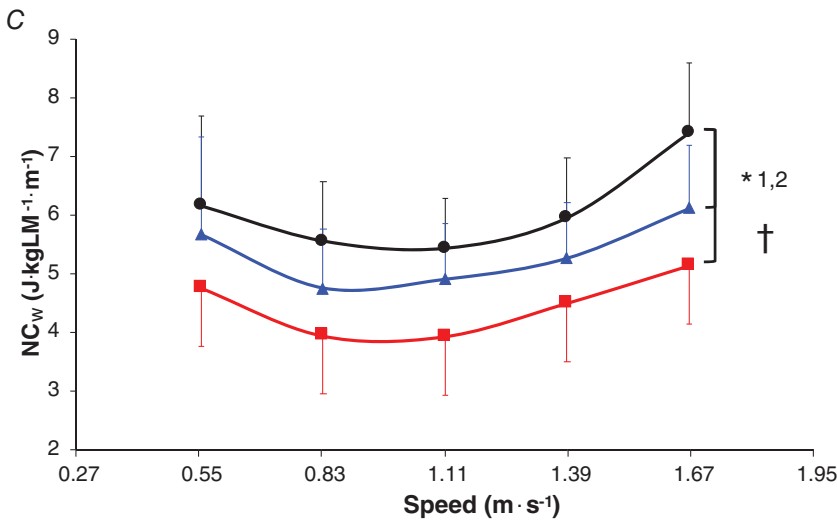

**Figure 1. Absolute net energy cost of walking (NC$_W$) (*A*), relative NC$_W$ per kg of body mass (*B*), and relative NC$_W$ per kg of lean mass (*C*) as a function of the walking speed**

Values are the mean ± SD. $n = 9$, except for 6 km h$^{-1}$: $n = 5$ at baseline, $n = 6$ at post 1 and $n = 8$ at post 2. Baseline, prebariatric surgery; post 1, −25% of initial body mass (∼6.6 months average) after bariatric surgery; post 2, 12 months after bariatric surgery. Black lines correspond to baseline, blue lines correspond to post 1 and red lines correspond to post 2. *1 indicates a significant difference ($P < 0.05$) between baseline and post 1. *2 indicates a significant difference ($P < 0.05$) between baseline and post 2. † indicates a significant difference ($P < 0.05$) between post 1 and post 2.

**Table 2. Changes in spatiotemporal parameters after bariatric surgery**

| | Step length (m)*[1,2] | Step duration (s)*[1,2] | Single support duration (s)*[1,2] | Double support duration (s)*[2] | Vertical displacements of CM (cm)*[2] |
|---|---|---|---|---|---|
| **0.56 m s⁻¹** | | | | | |
| Baseline | 0.42 ± 0.04 | 0.77 ± 0.08 | 0.46 ± 0.05 | 0.31 ± 0.04 | 2.1 ± 0.3 |
| Post 1 | 0.45 ± 0.03 | 0.81 ± 0.06 | 0.51 ± 0.05 | 0.30 ± 0.05 | 2.2 ± 0.2 |
| Post 2 | 0.45 ± 0.02 | 0.81 ± 0.04 | 0.52 ± 0.04 | 0.30 ± 0.05 | 2.2 ± 0.2 |
| **0.83 m s⁻¹** | | | | | |
| Baseline | 0.53 ± 0.03± | 0.64 ± 0.04 | 0.41 ± 0.04 | 0.23 ± 0.03 | 3.0 ± 0.5 |
| Post 1 | 0.56 ± 0.03 | 0.67 ± 0.03 | 0.46 ± 0.03 | 0.21 ± 0.03 | 3.1 ± 0.2 |
| Post 2 | 0.56 ± 0.03 | 0.67 ± 0.04 | 0.46 ± 0.02 | 0.21 ± 0.03 | 3.0 ± 0.3 |
| **1.11 m s⁻¹** | | | | | |
| Baseline | 0.63 ± 0.03 | 0.57 ± 0.03 | 0.38 ± 0.03 | 0.19 ± 0.02 | 4.0 ± 0.5 |
| Post 1 | 0.64 ± 0.02 | 0.58 ± 0.02 | 0.45 ± 0.08 | 0.19 ± 0.07 | 3.9 ± 0.5 |
| Post 2 | 0.65 ± 0.02 | 0.58 ± 0.02 | 0.42 ± 0.01 | 0.16 ± 0.02 | 3.8 ± 0.4 |
| **1.39 m s⁻¹** | | | | | |
| Baseline | 0.71 ± 0.03 | 0.51 ± 0.02 | 0.36 ± 0.02 | 0.15 ± 0.01 | 4.7 ± 0.5 |
| Post 1 | 0.73 ± 0.02 | 0.52 ± 0.01 | 0.40 ± 0.01 | 0.13 ± 0.01 | 4.6 ± 0.4 |
| Post 2 | 0.73 ± 0.02 | 0.53 ± 0.01 | 0.40 ± 0.02 | 0.13 ± 0.01 | 4.5 ± 0.4 |
| **1.67 m s⁻¹** | | | | | |
| Baseline | 0.79 ± 0.03 | 0.48 ± 0.02 | 0.35 ± 0.02 | 0.13 ± 0.01 | 5.5 ± 0.5 |
| Post 1 | 0.81 ± 0.03 | 0.48 ± 0.02 | 0.38 ± 0.02 | 0.10 ± 0.01 | 5.1 ± 0.4 |
| Post 2 | 0.80 ± 0.04 | 0.48 ± 0.02 | 0.38 ± 0.03 | 0.11 ± 0.01 | 4.9 ± 0.6 |

Values are the mean ± SD ($n = 9$, except for 6 km h⁻¹: $n = 5$ at baseline, $n = 6$ at post 1 and $n = 8$ at post 2). Baseline, pre-bariatric surgery; post 1, −25% of initial body mass (∼6.6 months average) after the bariatric surgery; post 2, 12-months after the bariatric surgery. *[1]indicates a significant difference ($P < 0.05$) between baseline and post 1. *[2]indicates a significant difference ($P < 0.05$) between baseline and post 2.

The absolute $NC_w$ significantly decreased by 23.5 ± 2.9% (averaged values across all speeds) between baseline and post 1 ($P < 0.0001$), 36.2 ± 1.6% between baseline and post 2 ($P < 0.0001$) and 16.3 ± 1.5% between post 1 and post 2 ($P < 0.0001$; Fig. 1A and Table S2). No significant difference was found in the relative $NC_w$ per kg of body mass between baseline and post 1 ($P = 1$), whereas a significantly decreased relative $NC_w$ was observed at post 2 compared to that at baseline and post 1 (−6.2 ± 2.7% and −8.1 ± 1.9%, respectively; $P \leq 0.007$; Fig. 1B). The mixed linear model for the absolute $NC_w$ with body mass as covariate ($P = 0.003$) confirmed these results with a significantly decreased absolute $NC_w$ at post 2 compared to that at baseline (mean difference: −19%; $P = 0.045$) and post 1 (mean difference: −11%; $P < 0.0001$) while there was no significant difference between baseline and post 1 ($P = 0.49$). The relative $NC_w$ per kg of lean mass significantly decreased between baseline and the two follow-ups (post 1: −12.1 ± 3.7% and post 2: −26.1 ± 2.1%; $P < 0.0001$ for both) and between post 1 and post 2 (−17.8 ±2.5%; $P = 0.033$; Fig. 1C). These results were confirmed by the mixed linear mixed model for the absolute $NC_w$ with lean body mass as covariate indicating that the absolute $NC_w$ significantly decrease between baseline and post 1 (mean difference:

−21%; $P < 0.0001$) and post 2 (mean difference: −35%; $P < 0.0001$) and between the two follow-ups (mean difference: −17%; $P < 0.0001$ for both).

For the energetic analyses, the mixed linear model showed no significant interaction effect ($P \geq 0.35$).

A positive and significant correlation was found between the difference in absolute $NC_w$ (J m⁻¹; averaged values across all speeds) and the change in total body mass between baseline and post 1 ($r = 0.71$; $P = 0.03$), whereas this correlation was not found between post 1 and post 2.

There was no significant correlation between the mass-normalized $NC_w$ (averaged values across all speeds) and the change in the mass-normalized SMR between baseline and post 2 ($r = 0.17$; $P = 0.66$).

## Mechanics

**Spatiotemporal parameters and kinematics.** The mixed linear model revealed for all spatiotemporal parameters a significant main time effect with no significant interaction effect ($P \geq 0.26$; Tables 2 and S3). Step duration (+3.2 ± 1.7% and +3.9 ± 2.2%) and length (+4.2 ± 1.6% and +4.8 ± 2.0%) and single support duration (+11.9 ± 2.9% and +10.5 ± 1.5%) significantly

**Table 3. Changes in kinematics after bariatric surgery**

| | Hip | | Knee | | | | Ankle |
|---|---|---|---|---|---|---|---|
| | Average AP (stance) (°)[*1] | Average AP (swing) (°)[*1†] | Average AP (stance) (°)[*1,2] | Flexion at the heel strike (°)[*1,2] | Maximal flexion (early stance) (°)[*1,2] | Delta ROM (early stance) (°) | Average AP (stance) (°) |
| **0.56 m s⁻¹** | | | | | | | |
| Baseline | −2.4 ± 6.8 | 24.6 ± 6.5 | 13.9 ± 4.5 | 11.3 ± 7.1 | 19.3 ± 5.9 | 8.0 ± 4.2 | 2.3 ± 4.0 |
| Post 1 | 0.6 ± 6.7 | 28.3 ± 5.6 | 13.6 ± 4.8 | 9.4 ± 5.8 | 18.7 ± 3.3 | 9.2 ± 5.0 | 4.1 ± 3.8 |
| Post 2 | -0.7 ± 5.0 | 27.0 ± 4.4 | 11.8 ± 3.3 | 7.6 ± 3.7 | 16.3 ± 3.0 | 8.7 ± 3.4 | 4.2 ± 2.6 |
| **0.83 m s⁻¹** | | | | | | | |
| Baseline | -0.6 ± 6.4 | 26.5 ± 3.5 | 15.8 ± 4.5 | 8.5 ± 4.7 | 21.9 ± 5.8 | 12.2 ± 3.1 | 3.7 ± 7.0 |
| Post 1 | 0.4 ± 6.5 | 30.0 ± 6.8 | 13.3 ± 3.6 | 6.2 ± 2.8 | 18.7 ± 3.5 | 12.6 ± 3.0 | 4.7 ± 4.1 |
| Post 2 | -0.9 ± 5.0 | 27.7 ± 5.9 | 13.3 ± 3.9 | 5.7 ± 2.8 | 17.0 ± 3.2 | 11.3 ± 2.3 | 4.8 ± 2.9 |
| **1.11 m s⁻¹** | | | | | | | |
| Baseline | -0.4 ± 6.1 | 27.3 ± 3.2 | 17.8 ± 4.6 | 8.7 ± 4.5 | 23.6 ± 5.5 | 14.2 ± 3.1 | 3.5 ± 4.1 |
| Post 1 | 1.3 ± 5.9 | 32.2 ± 5.9 | 13.9 ± 3.3 | 5.5 ± 2.4 | 19.6 ± 3.6 | 14.1 ± 2.4 | 6.5 ± 7.3 |
| Post 2 | -0.1 ± 4.1 | 30.3 ± 6.4 | 13.3 ± 3.7 | 5.5 ± 2.8 | 18.2 ± 3.3 | 12.5 ± 3.0 | 5.1 ± 2.8 |
| **1.39 m s⁻¹** | | | | | | | |
| Baseline | 0.2 ± 7.0 | 30.6 ± 4.9 | 17.0 ± 4.7 | 9.7 ± 5.8 | 24.5 ± 4.3 | 14.8 ± 4.1 | 3.9 ± 4.1 |
| Post 1 | 3.3 ± 7.0 | 33.1 ± 3.3 | 15.1 ± 2.8 | 6.3 ± 3.5 | 21.4 ± 4.6 | 15.1 ± 3.5 | 6.1 ± 4.4 |
| Post 2 | 2.8 ± 4.5 | 30.1 ± 3.4 | 15.6 ± 4.8 | 8.3 ± 7.5 | 21.4 ± 6.8 | 13.0 ± 3.7 | 5.9 ± 2.6 |
| **1.67 m s⁻¹** | | | | | | | |
| Baseline | 1.3 ± 7.7 | 29.3 ± 5.1 | 16.9 ± 4.0 | 9.9 ± 4.7 | 26.0 ± 4.6 | 16.7 ± 2.3 | 5.9 ± 3.6 |
| Post 1 | 7.7 ± 4.2 | 35.4 ± 4.2 | 15.6 ± 2.7 | 6.9 ± 2.9 | 23.0 ± 3.7 | 16.1 ± 4.3 | 4.7 ± 9.2 |
| Post 2 | 2.3 ± 4.3 | 33.9 ± 5.2 | 15.7 ± 4.3 | 8.4 ± 6.8 | 23.4 ± 3.6 | 15.0 ± 3.8 | 6.8 ± 2.8 |

Values are the mean ± SD ($n = 9$, except for 6 km h⁻¹: $n = 5$ at baseline, $n = 6$ at post 1 and $n = 8$ at post 2). Baseline, prebariatric surgery; post 1, −25% of initial body mass (∼6.6 months average) after bariatric surgery; post 2, 12 months after bariatric surgery; average AP, average angular position in the stance phase or swing phase; early stance, 15% of the stance phase duration; ROM, range of motion; Delta, difference between maximal knee flexion during early stance and knee flexion at the heel strike. *1 indicates a significant difference ($P < 0.05$) between baseline and post 1. *2 indicates a significant difference ($P < 0.05$) between baseline and post 2. † indicates a significant difference ($P < 0.05$) between post 1 and post 2.

increased between the baseline and two follow-ups ($P \leq 0.003$) with no significant differences between the two follow-ups ($P = 1$ for all; Table 2). In contrast, compared with those at baseline, the double support duration and vertical displacements of CM decreased during the follow-up but significantly only at post 2 ($-11.7 \pm 6.5\%$ and $-2.0 \pm 5.4\%$, respectively; $P \leq 0.02$) with no significant differences between post 1 and post 2 ($P \geq 0.31$; Table 2).

Table 3 shows that the average hip joint angular position in the stance phase significantly increased only between baseline and post 1 ($P = 0.001$), with no significant changes between the two follow-ups ($P = 0.24$). The average hip joint angular position in the swing phase significantly increased at post 1 compared with baseline ($P < 0.0001$) and then significantly decreased between the two follow-ups ($P = 0.014$; Tables 3 and S4). The average knee joint angular position in the stance phase, knee flexion at the heel strike and maximal knee flexion in the early stance phase significantly decreased between baseline and the two follow-ups ($P \leq 0.04$ and $P \leq 0.017$

for post 1 and post 2, respectively), with no significant difference between post 1 and post 2 ($P \geq 0.67$; Tables 3 and S4). There was no significant difference in the delta knee ROM in the early stance phase or the average ankle joint angular position in the stance phase ($P = 0.14$ and $P = 0.12$, respectively; Tables 3 and S4). For the kinematic analyses, the mixed linear model did not report any significant interaction effects ($P \geq 0.65$).

**Mechanical works and pendular energy-saving recovery.** In absolute values (J m⁻¹), $W_{ext}$, $W_k$, $W_p$ and $W_{tot}$ significantly decreased between baseline and post 1 ($-21.5 \pm 3.3\%$, $-24.0 \pm 2.8\%$, $-28.4 \pm 2.3\%$ and $-26.8 \pm 1.7\%$, respectively; $P < 0.0001$ for all) and post 2 ($-27.1 \pm 2.6\%$, $-30.7 \pm 4.2\%$, $-34.2 \pm 1.5\%$, and $-33.0 \pm 1.2\%$, respectively; $P < 0.0001$ for all) and between post 1 and post 2 ($-6.9 \pm 1.0\%$, $-10.0 \pm 1.8\%$, $-10.1 \pm 1.3\%$, and $-8.9 \pm 1.4\%$, respectively; $P \leq 0.008$ for both; Figs 2A, 3A, 3C and 2E and Table S5). Compared to baseline, the absolute $W_{int}$ was significantly decreased

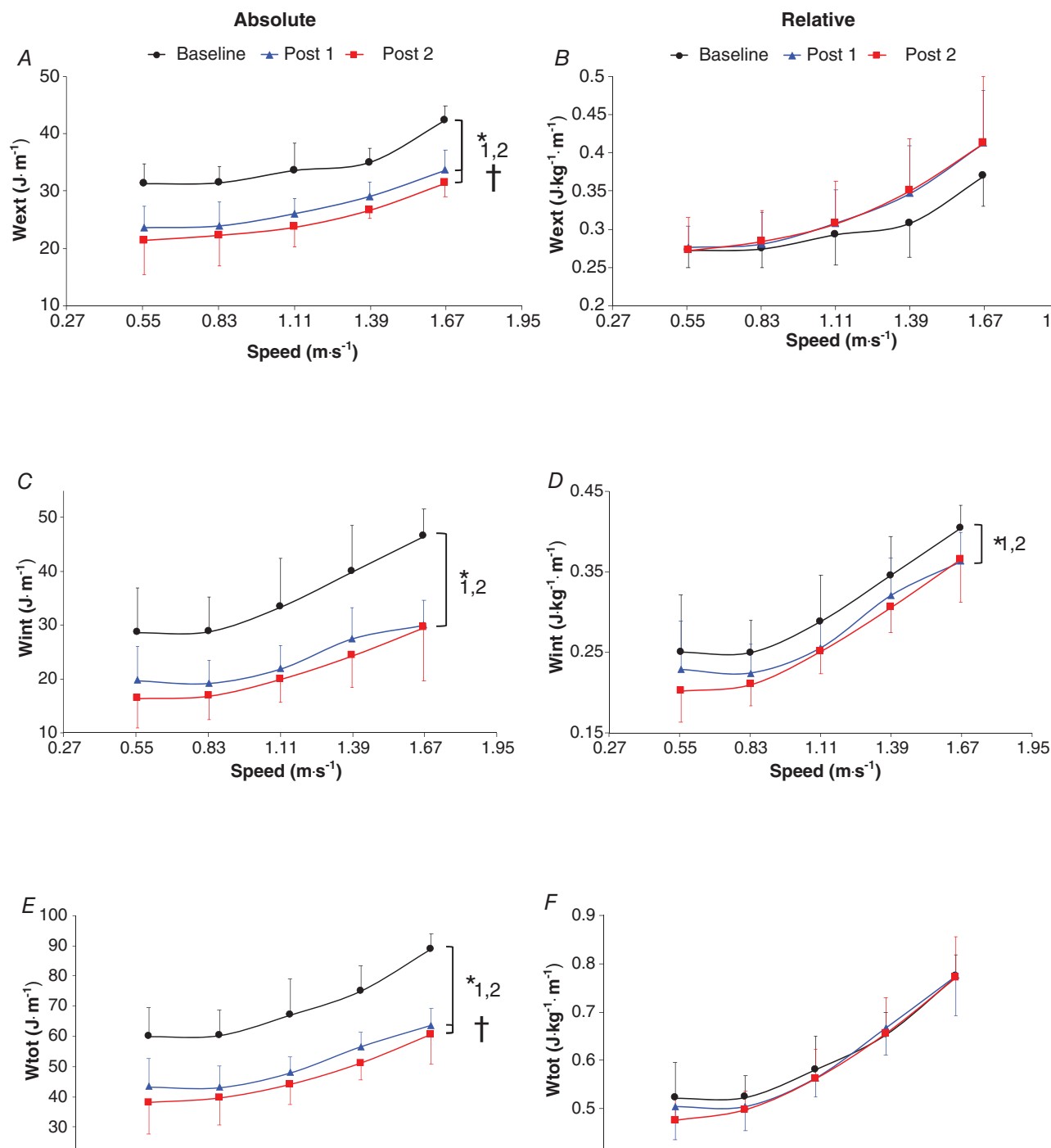

**Figure 2. Absolute external mechanical work ($W_{ext}$) (*A*), relative $W_{ext}$ per kg of body mass (*B*), absolute internal mechanical work ($W_{int}$) (*C*), relative $W_{int}$ per kg of body mass (*D*), absolute total mechanical work ($W_{tot}$) (*E*), and relative $W_{tot}$ per kg of body mass (*F*) as a function of walking speed**
Values are the mean ± SD. $n = 9$, except for 6 km h$^{-1}$: $n = 5$ at baseline, $n = 6$ at post 1 and $n = 8$ at post 2. Baseline, prebariatric surgery; post 1, −25% of initial body mass (∼6.6 months average) after bariatric surgery; post 2, 12 months after bariatric surgery. Black lines correspond to baseline, blue lines correspond to post 1 and red lines correspond to post 2. *1 indicates a significant difference ($P < 0.05$) between baseline and post 1. *2 indicates a significant difference ($P < 0.05$) between baseline and post 2. † indicates a significant difference ($P < 0.05$) between post 1 and post 2.

at post 1 ($-31.6 \pm 1.7\%$; $P < 0.0001$) and post 2 ($-38.6 \pm 1.2\%$; $P < 0.0001$), with no significant difference between the two follow-ups ($P = 0.083$; Fig. 2C and Table S5).

A trend towards a higher relative $W_{ext}$ per kg of body mass was found after bariatric surgery ($+5.9 \pm 4.6\%$ and $6.7 \pm 5.7\%$, respectively; $P \leq 0.09$), with no significant difference between post 1 and post 2 ($P > 0.1$; Fig. 2B). The relative $W_{int}$ and $W_p$ were significantly lower at post 1 ($-7.9 \pm 2.2\%$ and $-3.6 \pm 3.3\%$, respectively) and post 2 ($-11.4 \pm 1.4\%$ and $-5.1 \pm 3.2\%$, respectively) than at baseline ($P \leq 0.02$ and $P \leq 0.008$, respectively), with similar values found between the two follow-ups ($P = 0.91$ and $P = 0.21$, respectively; Figs 2D and 3D). No significant difference was found in the relative $W_{tot}$ and $W_k$ after

bariatric surgery ($P = 0.36$ and $P = 0.07$, respectively; Figs 2F and 3B).

A significantly lower recovery was found at post 1 than at baseline ($-3.0 \pm 1.5\%$; $P = 0.013$) and at post 2 than at baseline ($-4.6 \pm 1.1\%$; $P < 0.0001$), with no significant difference between post 1 and post 2 ($P = 0.64$; Fig. 4A and Table S5).

For all these analyses, the mixed linear model showed no significant interaction effect ($P \geq 0.60$).

## Mechanical efficiency

Mechanical efficiency (main time effect: $P = 0.016$ and interaction effect: $P = 0.82$) was similar at post 1 and post 2 when compared to that at baseline ($P \geq 0.19$), but it was

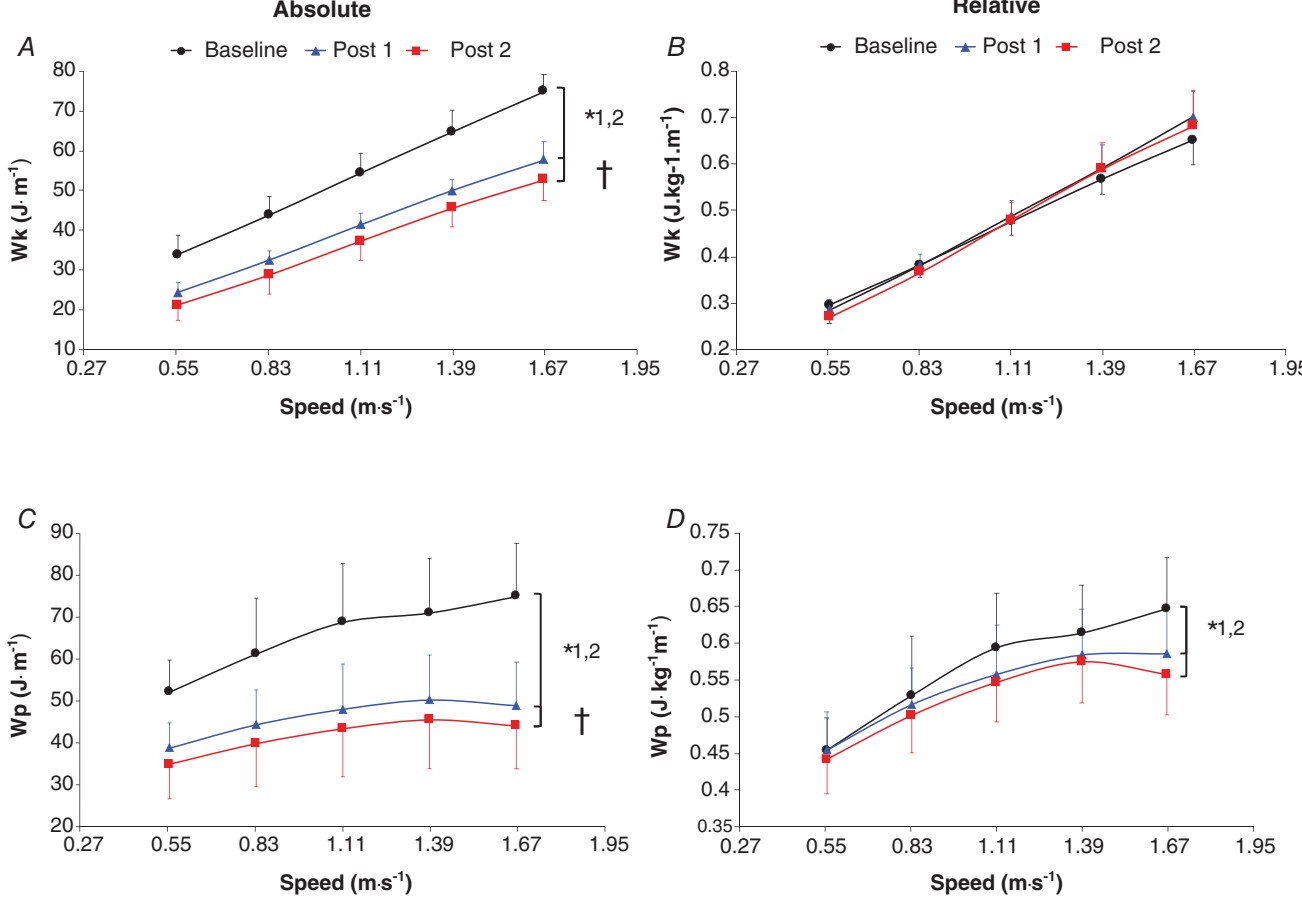

**Figure 3. Absolute kinetic mechanical work ($W_K$) (A), relative $W_k$ per kg of body mass (B), absolute potential mechanical work ($W_p$) (C), and relative $W_p$ per kg of body mass (D) as a function of walking speed**

Values are the mean $\pm$ SD. $n = 9$, except for 6 km h$^{-1}$: $n = 5$ at baseline, $n = 6$ at post 1 and $n = 8$ at post 2. Baseline, prebariatric surgery; post 1, $-25\%$ of initial body mass ($\sim$6.6 months average) after bariatric surgery; post 2, 12 months after bariatric surgery. Black lines correspond to baseline, blue lines correspond to post 1 and red lines correspond to post 2. *1 indicates a significant difference ($P < 0.05$) between baseline and post 1. *2 indicates a significant difference ($P < 0.05$) between baseline and post 2. † indicates a significant difference ($P < 0.05$) between post 1 and post 2.

significantly higher at post 2 than at post 1 ($+14.1 \pm 4.6\%$; $P = 0.013$; Fig. 4$B$ and Table S5).

## Discussion

The main finding of this study was that bariatric surgery significantly decreased the mass-normalized energy cost of walking only after 1 year (post 2) with similar relative total mechanical work and, thus, with an increased mechanical efficiency. This partially confirms our hypothesis because the mechanical and energetic adaptations to body mass loss were time dissociated during the 1 year follow-up. The former emerged quickly and was a mainly mass-driven adaptation, whereas the latter appeared only after a period of adaptation to the new gait pattern as a more adaptive and behavioural change.

The results of this study showed a significant decrease in the total body mass at post 1 and post 2 after bariatric surgery ($-25.7 \pm 3.4\%$ and $-31.9 \pm 8.1\%$, respectively), inducing a significant decrease in BMI at both time points. During the 1-year follow-up, the class of obesity of the participants, determined by the averaged BMI, changed from class III to overweight (Table 1). Fat body mass loss contributed more than lean body mass loss to the reduction in total body mass ($-19\%$ and $-6\%$ at post 1 and $-25\%$ and $-6\%$ at post 2, respectively). This was also due to a significant decrease in lean body fat mass only at post 1, with no additional reduction at post 2 compared to post 1 (Table 1), corroborating previous findings (van Gemert *et al.* 1998; Browning *et al.* 2016). Similar loss changes during the follow-up characterized the mass, fat and lean mass of the limbs and the mass and lean mass of the trunk (Table 1). In contrast, fat trunk mass significantly decreased at post 1 and post 2, with an additional significant decrease between post 1 and post 2, totally explaining the reduction in fat and total body mass between these two time points (Table 1). The decreased fat trunk mass was accompanied by an important reduction in visceral adipose tissue at post 1 and post 2, commonly associated with several

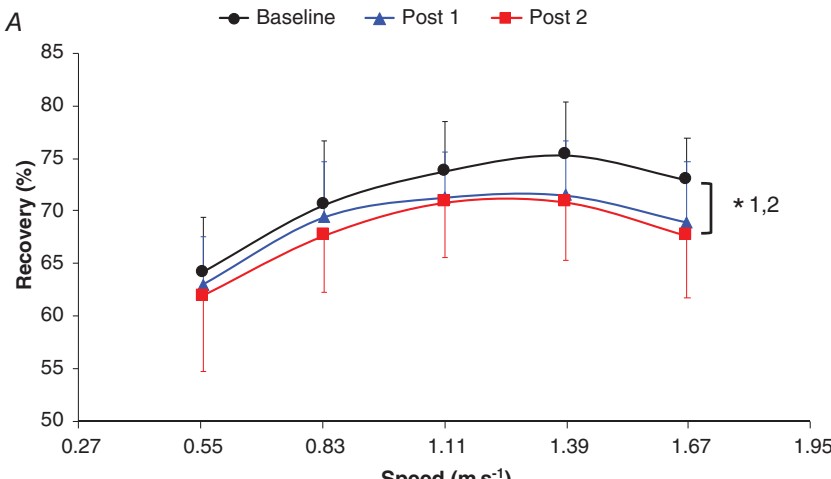

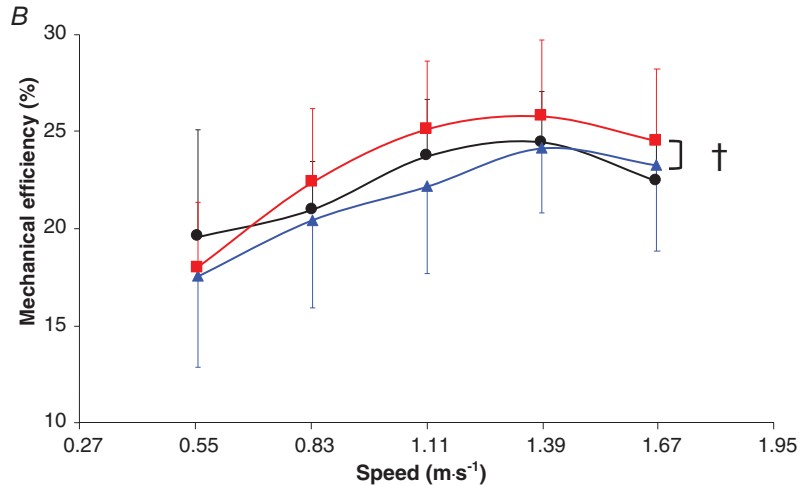

**Figure 4. Recovery (*A*) and mechanical efficiency (*B*) as a function of walking speed**
Values are the mean $\pm$ SD. $n = 9$, except for 6 km h$^{-1}$: $n = 5$ at baseline, $n = 6$ at post 1 and $n = 8$ at post 2. Baseline, prebariatric surgery; post 1, $-25\%$ of initial body mass ($\sim$6.6 months average) after bariatric surgery; post 2, 12 months after bariatric surgery. Black lines correspond to baseline, blue lines correspond to post 1, and red lines correspond to post 2. *1, indicating a significant difference ($P < 0.05$) between baseline and post 1. *2 indicates a significant difference ($P < 0.05$) between baseline and post 2. † indicates a significant difference ($P < 0.05$) between post 1 and post 2.

cardiometabolic risk factors (Elffers *et al.* 2017), and attesting to the positive effect of bariatric surgery. This was obtained with no significant change in physical activity level during follow-up, confirming previous results (for review see Li *et al.* 2019).

This very large body mass loss also induced gait functional improvements, with an increase in PWS found at post 1 ($+23.5 \pm 15.4\%$) and post 2 ($+24.7 \pm 13.9\%$) with no significant difference between these time points. The baseline values ($1.07$ m s$^{-1}$) are in line with those of previous studies (Spyropoulos *et al.* 1991; Mattsson *et al.* 1997; Lai *et al.* 2008; Malatesta *et al.* 2009) but slower than those of others (Browning *et al.* 2006; Fernandez Menendez *et al.* 2019*a*). The PWS difference among the studies might be due to the methodological differences (overground *vs.* treadmill; Malatesta *et al.* 2017) and/or obesity class of the participants of these studies (Browning *et al.* 2006). Adults with obesity prefer to walk at a slower walking speed than their lean counterparts (Spyropoulos *et al.* 1991; McGraw *et al.* 2000; Malatesta *et al.* 2009) as a function of several biomechanical and energetic competing demands (Fernandez Menendez *et al.* 2019*a*). This slower PWS was associated with a shorter step length and a lower step frequency, increasing the time spent during double support (Spyropoulos *et al.* 1991; DeVita & Hortobagyi, 2003; Browning *et al.* 2006; Malatesta *et al.* 2009). At the five fixed walking speeds used in the present study, our findings showed that body mass loss significantly increased step length and duration and single support duration, directly and indirectly confirming that bariatric surgery may reverse the spatiotemporal changes induced by obesity at fixed walking speed and PWS, respectively. The increased PWS already at post 1 corroborated the suggestion of Hortobagyi *et al.* (2011) that after the body mass limiting gait changes threshold ($-30$ kg or 25% of the initial of body mass as that obtained in post 1), body mass loss may produce both mass-driven and adaptive adaptations in gait behaviour (i.e. concept of mechanical plasticity of human gait).

This study is the first to investigate the effect of walking mechanical plasticity induced by large body mass loss after bariatric surgery on the energetics and efficiency of walking at several speeds. Our findings showed that the absolute NC$_w$ (J m$^{-1}$) decreased by $23.5 \pm 2.9\%$ at post 1 and in the same proportion of the total body mass loss at the same time point ($-25.7 \pm 3.4\%$; Fig. 1*A*). However, NCw was reduced by $36.2 \pm 1.6\%$ between baseline and post 2 and more than $-31.9 \pm 8.1\%$ of total body mass loss in the same period of follow-up. Moreover, the correlation found between the difference in absolute NC$_w$ and the change in total body mass only between baseline and post 1, where the majority of the body mass was lost, suggests that body mass loss, not the amount of fat mass loss, is the main factor involved in the improvement of the economy of walking in individuals with obesity

after bariatric surgery. However, a significant reduction in the relative NC$_w$ (J kg$^{-1}$ m$^{-1}$) was observed only at post 2 (Fig. 1*B*), attesting that other factors rather than only body mass have an influence on the energetics of walking (Browning *et al.* 2006; Peyrot *et al.* 2009, 2010). The increase in the relative SMR (W kg$^{-1}$) at post 2 (Table 1), likely due to the substantial reduction in the metabolically inactive fat body mass, despite the loss of the total lean body mass, may partially contribute to the significant reduction in the mass-normalized NC$_w$ only at post 2 (Browning *et al.* 2016). Although an increase in the relative SMR after a non-surgical intervention (Peyrot *et al.* 2010) and in relative resting metabolic rate after bariatric surgery (de Cleva *et al.* 2018) has already been reported, the comparison of these mass-normalized measures of standing or resting metabolic rate (simple ratio-based assessments) before and after body mass loss is questioned (Cooper & Berman, 1994; Packard & Boardman, 1999; Browning *et al.* 2018). In fact, using body mass as covariate, we did not find any differences in the absolute SMR during follow-up (Table 1), attesting that the ratio might introduce problems with respect to statistical analysis and interpretation of SMR data (Browning *et al.* 2018). However, there was no correlation between the difference in the mass-normalized SMR and the change in the relative NC$_w$ (J kg$^{-1}$ m$^{-1}$) between baseline and post 2. Moreover, compared to the baseline, the relative SMR per kg of lean body mass decreased only at post 1 and then returned to baseline levels at post 2 (Table 1). This could indicate that 'metabolic adaptation' (Rosenbaum & Leibel, 2010) in response to weight loss could be greater in post 1, and that this effect weakened as participants' bodies adjusted to their new body mass and composition, confirming previous findings after bariatric surgery (Knuth *et al.* 2014). This indirectly corroborates that the increase in the mass-normalized SMR at post 2 may not be the main determinant in the decrease in the mass-normalized NC$_w$ at this time point. Besides, the results of the mixed linear model for the absolute NC$_w$ with body mass as covariate confirmed those of the mass-normalized NC$_w$ analysis (ratio analysis). This demonstrates that, for the main outcome of the study (the relative NC$_w$), (i) the ratio analysis can be correctly used to analyse our data (Packard & Boardman, 1999) and (ii) the decrease in body mass is not the only factor involved in the reduced relative NC$_w$ after bariatric surgery. Another likely explanatory factor may be the mechanical gait changes during the follow-up.

However, no change was observed in the mass-normalized $W_{tot}$ after bariatric surgery (Fig. 2*F*). The unexpected invariability in the relative $W_{tot}$ was caused by compensation between the mass-normalized $W_{ext}$ and $W_{int}$ (Figs 2*B* and *D*). $W_{ext}$ tended to increase during the first time point and then remained constant. This was caused by the significantly lower recovery

found after bariatric surgery (Fig. 4*A*), corroborating our previous findings reporting a skilful transduction of mechanical energy in adults with obesity used to minimize the amount of mass-normalized $W_{ext}$ performed during walking (Fernandez Menendez *et al.* 2019*b*, 2020). This improved pendular energy transduction (i.e. mechanical plasticity) seems to be involved in the response to precocious and chronic adaptation to loading in adults with Prader–Willi syndrome developing morbid obesity during early childhood (Malatesta *et al.* 2013). This was obtained by increasing the mass-normalized $W_p$ to optimize the relative magnitude of kinetic and potential energy during walking at PWS. The same adaptation, but in the opposite direction, appears to be involved after large body mass loss with a decreased mass-normalized $W_p$ and vertical displacements of CM associated with similar mass-normalized $W_k$ during the follow-up compared to baseline (see Fig. 3 and Table 2). This less bouncing walking was associated with pivotal changes in knee ROM and joint angular position during stance: a decreased average knee joint angular position in the stance phase, knee flexion at the heel strike and maximal knee flexion in the early stance phase at fixed walking speed at post 1 and post 2 (Table 3). Our results are in contrast with those of Hortobagyi *et al.* (2011), who showed that after bariatric surgery-induced very large body mass loss (−34%), the gait pattern became more dynamic with increased knee flexion at faster PWS. However, this knee adaptation may be due to the increased PWS after bariatric surgery and not related to body mass loss. In fact, knee flexion during early stance increases as a function of walking speed (Willems & Schepens, 2012), and the same authors did not find the same knee adaptations at fixed walking speed (1.5 m s$^{-1}$) after surgery (Hortobagyi *et al.* 2011). In contrast, our results indirectly corroborate those showing increased knee flexion at heel strike and greater knee flexion in repose to loading as a 'protective measure' to improve the absorption of load during the foot strike and to improve posture stability during walking as compensatory reflex adaptations (Fouad *et al.* 2001; Attwells *et al.* 2006; Majumdar *et al.* 2010). These loading adaptations are lost (or significantly decreased) after a very large body mass loss induced by bariatric surgery in our participants, attesting that these kinematic adaptations and pendulum-like mechanisms are reversible, as previously shown in African women (Heglund *et al.* 1995). The less knee flexed lower limbs at post 1 than at baseline may induce advantageous joint moments associated with lower muscle activation (Grabowski *et al.* 2005; Ortega & Farley, 2005) and $NC_w$ (Griffin *et al.* 2003). However, the increased $W_{ext}$, which accounts for about one-half of $NC_w$ in adults (Grabowski *et al.* 2005), at post 1 compared to baseline may contribute to blunt this expected improvement in walking economy maintaining similar $NC_w$ between these two time points.

Interestingly, while the relative $W_{ext}$ tended to increase, the mass-normalized $W_{int}$ behaved in the opposite direction, with a decreased value at post 1 compared with that at baseline, which remained constant afterwards and closely related to the body mass loss changes during the follow-up (Fig. 2*D*). Therefore, the amount of mass-normalized $W_{tot}$ was not altered after the surgery (Fig. 2*F*).

These findings suggest that after a very large body mass loss, individuals with obesity may readapt their walking pattern towards a gait similar to that of normal body weight adults to maintain a constant amount of total work performed during walking and thereby control the energy consumption. Moreover, it seems that while mechanical adaptations are relatively quickly developed after body mass loss (i.e. mass-driven changes), the energetic aspect needs a period of adjustment to the new gait pattern to make the muscles work in an efficient manner (Massaad *et al.* 2007). Corroborating this hypothesis, our results showed a significant increase in mechanical efficiency only at post 2, due to the similar mass-normalized $W_{tot}$ and decrease in relative $NC_w$ at this time point compared with those at post 1. This improved mechanical efficiency may be due to an enhancement in muscle contraction efficiency but also an increase in propulsive efficiency (i.e. the transformation of the positive work performed by muscle to mechanical work) (the two components of overall efficiency of walking; Cavagna, 2017). In fact, at post 1, mechanical efficiency was similar to the baseline values (no significant changes in mass-normalized $NC_w$ and $W_{tot}$ at this time point) and associated with significantly decreased $NC_w$ normalized by lean body mass (Fig. 1*C*), essentially and proportionally due to a decrease in lean body mass at post 1 *vs.* post 2 (Table 1). From post 1 to post 2, no further decrease in lean body mass was associated with a disproportional and significant decrease in lean mass-normalized $NC_w$ (−17.8 ± 2.5%), indirectly attesting to an improvement in muscle contraction efficiency. Moreover, the 'mechanical stability' of the gait pattern, between post 1 and post 2, would be a period of adjustment to the new gait pattern, developed from baseline to post 1 and essentially due to the very large body mass loss, necessary to an enhancement in propulsive efficiency. This improvement, obtained during walking with the new gait pattern in the second part of the follow-up (i.e. adaptive and behavioural changes), may be due to lower muscle activations and by making the muscles work in more favourable conditions (e.g. decreased volume of active muscle operating at advantageous length and/or velocities). This also confirms that human walking optimization is a compromise between saving energy via pendulum-like mechanism and making muscles work efficiently (Massaad *et al.* 2007).

The enhanced mechanical efficiency at post 2 corroborates previous findings showing an improved

muscle metabolic economy and mitochondrial function after diet- (Newcomer *et al.* 2001) or bariatric surgery-induced (Nijhawan *et al.* 2013; Vijgen *et al.* 2013; Fernstrom *et al.* 2016) body mass loss. Neuro-endocrine changes (e.g. a decrease in triiodothyronine concentration, an increase in sympathetic tone and a reduction in parasympathetic tone) might enhance skeletal muscle contractile energy efficiency after body mass loss (Rosenbaum & Leibel, 2010; Galgani & Santos, 2016). However, some studies reported that the cycling efficiency (Rosenbaum *et al.* 2003) and $NC_w$ (Borges *et al.* 2018) were improved directly after shorter body mass loss interventions and, thus, somewhat in contrast with our delayed enhancement in $NC_w$ and mechanical efficiency only at post 2. This might reflect the different degree of body mass loss obtained in individuals with overweight by these previous studies ($-10$ to $-12$ kg corresponding to 10–16% of the initial body mass) compared with that of the present study at post 1 ($-30$ kg or $-25\%$ of the body mass at baseline). Moreover, this difference indirectly confirms that a very large body mass loss, inducing a greater degree of gait pattern change, would need a period of adaptation to the new gait pattern to improve its mechanical efficiency.

The main limitation of this study was the small sample size ($n = 9$). Each month for 2 years, the researchers presented the study design and protocol to potential participants during a session of the education programme performed before the bariatric surgery. Only 11 individuals agreed to participate in the protocol, and two of them dropped out after the baseline assessments for reasons independent of the study's protocol. This attests to the difficulty of recruiting and retaining participants, as previously highlighted in the same population by others (Hortobagyi *et al.* 2011).

In conclusion, our findings showed that after a very large body mass loss, individuals with obesity may reorganize their walking pattern into a gait more similar to that of adults of normal body mass, thus decreasing their energy cost of walking by making their muscles work more efficiently. However, and according to our results, while the mechanical modifications emerged quickly after the limiting gait changes threshold ($\sim25\%$ of the initial body mass) and were mainly mass-driven changes, the energetic improvement appeared after a period of adaptation to the new walking pattern (adaptive changes), when the muscles were able to provide mechanical work in more efficient conditions (improvement in propulsive and muscle contraction efficiency).

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

## Additional information

### Data availability statement

The datasets supporting this article are available on request to the corresponding author.

### Competing interests

The authors declare no conflict of interest.

### Author contributions

Conceptualization, D.M., M.S., L.F. and A.F.M.; methodology, D.M., J.F., B.U., D.H., M.S., L.F. and A.F.M.; investigation, D.M. and A.F.M.; formal analysis, D.M. and A.F.M.; writing – original draft preparation, D.M.; writing – review and editing, D.M., J.F., B.U., D.H., M.S., L.F. and A.F.M.; Supervision: D.M. and A.F.M. All authors have read and approved the final version of this manuscript and agree to be accountable for all aspects of the work in ensuring that questions related to the accuracy or integrity of any part of the work are appropriately investigated and resolved. All persons designated as authors qualify for authorship, and all those who qualify for authorship are listed.

### Funding

No funding was received for this work.

### Acknowledgements

The authors warmly thank the participants for their time and cooperation and Fabienne von Roten for statistical assistance.

### Keywords

bariatric surgery, economy, energy cost of walking, gait, locomotion, mechanical plasticity

## Supporting information

Additional supporting information can be found online in the Supporting Information section at the end of the HTML view of the article. Supporting information files available:

**Peer Review History**
**Statistical Summary Document**
**Supplementary Table S1**: Participant characteristics (raw data).
**Supplementary Table S2**: Standing metabolic rate and energetics of walking (raw data).
**Supplementary Table S3**: Spatiotemporal parameters and preferred walking speed (raw data).
**Supplementary Table S4**: Kinematics (raw data).
**Supplementary Table S5**: Mechanics and mechanical efficiency (raw data).

