## [Peer Review History · The Journal of Physiology]

Effect of very large body mass loss on energetics, mechanics and efficiency of walking in adults with obesity: mass-driven vs behavioural adaptations

Davide Malatesta, Julien Favre, Baptiste Ulrich, Didier Hans, Michel Suter, Lucie Favre, and Aitor Fernández Menéndez

DOI: 10.1113/JP281710

Corresponding author(s): Davide Malatesta (davide.malatesta@unil.ch)

Review Timeline:	Submission Date:	31-Mar-2021
	Editorial Decision:	11-May-2021
	Revision Received:	22-Jun-2021
	Editorial Decision:	12-Jul-2021
	Revision Received:	04-Aug-2021
	Accepted:	26-Aug-2021

Senior Editor: Michael Hogan

Reviewing Editor: Audrey Bergouignan

Transaction Report:

Dear Dr Malatesta,

Re: JP-RP-2021-281710 "Effect of very large body mass loss on energetics, mechanics and efficiency of walking in adults with obesity: mass-driven vs behavioural adaptations" by Davide Malatesta, Julien Favre, Baptiste Ulrich, Didier Hans, Michel Suter, Lucie Favre, and Aitor Fernández Menéndez

Thank you for submitting your manuscript to The Journal of Physiology. It has been assessed by a Reviewing Editor and by 2 expert Referees and I am pleased to tell you that it is considered to be acceptable for publication following satisfactory revision.

The reports are copied at the end of this email. Please address all of the points and incorporate all requested revisions, or explain in your Response to Referees why a change has not been made.

NEW POLICY: In order to improve the transparency of its peer review process The Journal of Physiology publishes online as supporting information the peer review history of all articles accepted for publication. Readers will have access to decision letters, including all Editors' comments and referee reports, for each version of the manuscript and any author responses to peer review comments. Referees can decide whether or not they wish to be named on the peer review history document.

I hope you will find the comments helpful and have no difficulty returning your revisions within 4 weeks.

Your revised manuscript should be submitted online using the links in Author Tasks Link Not Available.

Any image files uploaded with the previous version are retained on the system. Please ensure you replace or remove all files that have been revised.

REVISION CHECKLIST:

- Article file, including any tables and figure legends, must be in an editable format (eg Word)
- Upload each figure as a separate high quality file
- Upload a full Response to Referees, including a response to any Senior and Reviewing Editor Comments;
- Upload a copy of the manuscript with the changes highlighted.

- A potential 'Cover Art' file for consideration as the Issue's cover image;
- Appropriate Supporting Information (Video, audio or data set <https://jp.msubmit.net/cgi->

bin/main.plex?form_type=display_requirements#supp).

To create your 'Response to Referees' copy all the reports, including any comments from the Senior and Reviewing Editors, into a Word, or similar, file and respond to each point in colour or CAPITALS and upload this when you submit your revision.

I look forward to receiving your revised submission.

If you have any queries please reply to this email and staff will be happy to assist.

Yours sincerely,

Michael C. Hogan
Senior Editor
The Journal of Physiology
<https://jp.msubmit.net>
<http://jp.physoc.org>
The Physiological Society
Hodgkin Huxley House
30 Farringdon Lane
London, EC1R 3AW
UK
<http://www.physoc.org>
<http://journals.physoc.org>

REQUIRED ITEMS:

-Author photo and profile. First (or joint first) authors are asked to provide a short biography (no more than 100 words for one author or 150 words in total for joint first authors) and a portrait photograph. These should be uploaded and clearly labelled with the revised version of the manuscript. See Information for Authors for further details.

-The contact information provided for the person responsible for 'Research Governance' at your institution is an author on this paper. Please provide an alternative contact who is not an author on this paper or confirm that the author whose email was provided has sole responsibility for research governance. This is the person who is responsible for regulations, principles and standards of good practice in research carried out at the institution, for instance the ethical treatment of animals, the keeping of proper experimental records or the reporting of results.

-You must start the Methods section with a paragraph headed Ethical Approval. If experiments were conducted on humans confirmation that informed consent was obtained, preferably in writing, that the studies conformed to the standards set by the latest revision of the Declaration of Helsinki, and that the procedures were approved by a properly constituted ethics committee, which should be named, must be included in the article file. If the research study was registered (clause 35 of the Declaration of Helsinki) the registration database should be indicated, otherwise the lack of registration should be noted as an exception (e.g. The study conformed to the standards set by the Declaration of Helsinki, except for registration in a database.). For

further information see: <https://physoc.onlinelibrary.wiley.com/hub/human-experiments>

-Please upload separate high-quality figure files via the submission form.

-Your paper contains Supporting Information of a type that we no longer publish. Any information essential to an understanding of the paper must be included as part of the main manuscript and figures. The only Supporting Information that we publish are video and audio, 3D structures, program codes and large data files. Your revised paper will be returned to you if it does not adhere to our Supporting Information Guidelines

-A Statistical Summary Document, summarising the statistics presented in the manuscript, is required upon revision. It must be on the Journal's template, which can be downloaded from the link in the Statistical Summary Document section here: https://jp.msubmit.net/cgi-bin/main.plex?form_type=display_requirements#statistics

-Papers must comply with the Statistics Policy https://jp.msubmit.net/cgi-bin/main.plex?form_type=display_requirements#statistics

In summary:

-If n {less than or equal to} 30, all data points must be plotted in the figure in a way that reveals their range and distribution. A bar graph with data points overlaid, a box and whisker plot or a violin plot (preferably with data points included) are acceptable formats.

-If $n > 30$, then the entire raw dataset must be made available either as supporting information, or hosted on a not-for-profit repository e.g. FigShare, with access details provided in the manuscript.

-' n ' clearly defined (e.g. x cells from y slices in z animals) in the Methods. Authors should be mindful of pseudoreplication.

-All relevant ' n ' values must be clearly stated in the main text, figures and tables, and the Statistical Summary Document (required upon revision)

-The most appropriate summary statistic (e.g. mean or median and standard deviation) must be used. Standard Error of the Mean (SEM) alone is not permitted.

-Exact p values must be stated. Authors must not use 'greater than' or 'less than'. Exact p values must be stated to three significant figures even when 'no statistical significance' is claimed.

-Statistics Summary Document completed appropriately upon revision

EDITOR COMMENTS

Reviewing Editor:

The statistical summary document is missing.

Results from this study are interesting and adds significantly to the body of literature relevant to adaptive thermogenesis following weight loss. However, concerns raised by the two referees need to be addressed. I encourage you to carefully consider the comments related to the relationship between changes in body weight and changes in walking cost. As suggested, it is also important to normalize the SMR data for the changes in body weight/composition by using an ANCOVA and not by simply dividing W per kg. Also please address the questions of both Referees related to the increase in SMR at point 2. Finally, it is important to appropriately indicate the units of the study outcomes to have a clear and correct message. Thank you.

REFEREE COMMENTS

Referee #1:

The authors have examined the effect of substantial weight loss on the energy cost of walking and the mechanical work of walking by comparing these variables in participants with obesity at baseline and then at 2 time points post bariatric surgery. The hypotheses and methods are well-articulated and the work was conducted with appropriate protocols. The analyses are relatively straightforward. My concerns regarding the paper and analyses (and conclusions) are as follows:

1. The primary result of the paper is that there appears to be an improvement (decrease) in walking energy cost at post-2, which the authors attribute to increased efficiency. I'm not certain that this is the only interpretation of their data. The changes in walking cost as a % (-25% at post 1, -38% at post 2) match the change in weight (-26% post 1, -32% post 2) almost precisely. This suggests the change in weight is sufficient to explain the change in walking cost, and in fact there is no difference in net walking cost per kg in pre vs post-1. The slightly lower cost/kg at post-2 is completely dependent on the *increased* SMR at Post-2. Further, it is *highly* unusual to have an increased SMR (absolute) in subjects who have lost weight (as they do from post1 from post2). Therefore, I strongly encourage the authors to consider the possibility that lower walking cost at post-2 could be an artifact of greater SMR at post-2, which could itself be due to any number of factors: were the subjects truly at a stable SMR at the end of the 5-minute trial? (5 minutes is quite short for measuring resting expenditures). Were the subjects completely fasted for all trials? Were post1 and post2 performed at a similar time of day, with similar physical activity the preceding day? Is there any chance of measurement or machine error in Post 1 or 2? The effect would not have to be large to result in the reported decrease in walking cost at post-2.

2. If the authors can convincingly rule out that any issue with SMR at post2, the authors should

articulate a plausible mechanism for the improved efficiency at Post-2. The authors attribute this improved efficiency to a 'readapted' gait, but they have measured all of the relevant gait parameters and none of them seem to account for the improved ratio of metabolic energy / mechanical work at post-2. Indeed, recovery% drops at post1 and post2. Do the authors propose that the muscles themselves become more efficient? If not, which aspects of gait could be changing in a favorable manner to improve efficiency that aren't already captured and analyzed? Of course, if the energy measurements at post-2 are low because of an artifact in the measurement of SMR at post-2, then the apparent improvement in efficiency at post-2 will likely fall away, making this point moot (and substantially changing the framing of the Discussion).

3. Introduction, paragraph 1: It's true that weight gain must be due to energy imbalance, but the discussion here implies that weight gain and obesity result from lower energy expenditure. The available evidence points to overconsumption of calories as a much larger factor in the development of obesity than a decrease in expenditure. See, for example, work by Prentice, Dugas, Luke, or Pontzer showing that total daily energy expenditures are not lower in people with obesity and that total daily expenditures do not correspond with daily physical activity.

4. Introduction: Defining the net cost of walking here, may want to include that this is the energy expended over and above resting energy expenditure.

5. Introduction: Regarding the greater cost of walking with obesity and its effects on NEAT: it seems equally plausible that the greater walking costs (J / km) would result in greater NEAT with obesity, even if steps/day is reduced.

6. Methods: Please briefly review the protocol for determining preferred walking speed.

7. See comment 1 above: please provide more information on time of day, whether subjects were fasted, how SMR was established to be stable, etc.

8. Results: In reporting the SMR pre- and post-weight loss, please be aware that it's inappropriate to compare the simple ratios of energy/kg. Using this simple ratio assumes implicitly that the relationship between SMR and body mass (or lean mass) is a straight line with an intercept at $y=0$. That is very rarely the case with resting or total daily metabolic rates. The appropriate way to examine the relationship with body size is to use regression analysis or ANCOVA, with lean mass as the covariate and time point (pre, post1, post2) as a categorical variable. (you could add other variables to lean mass, but with a small sample size that's probably not advisable). [note: this may also affect the discussion of SMR changes in the Discussion page 21]. See: D. B. Allison, F. Paultre, M. I. Goran, E. T. Poehlman, S. B. Heymsfield, Statistical considerations regarding the use of ratios to adjust data. *Int J Obes Relat Metab Disord* 19, 644-652 (1995).

Referee #2:

General Comments:

This is a study of changes in energy expenditure by muscle during walking before and after bariatric surgery. The authors' main finding is that increased muscle efficiency, defined as work/kg, is not significantly increased until 1 year after bariatric surgery.

This is very interesting. The pre- and post- bariatric surgery studies and overall attention to detail are strengths, but the manuscript is difficult to follow. Energy expenditure is not directly expressed since data are generally presented with "work performed" rather than energy burned during exercise. Energy expenditure and work are interchanged in a very confusing manner. For example, W_{tot} is defined as total mechanical work, but standing metabolic rate, which should really be somewhat equivalent of a standing RMR, is also defined as work. Since no object is really moving during SMR - work seems to be an inappropriate measure. This terminology is used in other papers by this group but can be difficult to follow. Clarification would be helpful.

Abstract:

Throughout the manuscript please use "people first" language. For example, please use the term "individuals with obesity" rather than "obese individuals".

The first sentence is self-contradictory. Specifically, improving the higher energy cost of walking cannot result in lower energy expenditure and it is unlikely that a large loss of mass would somehow resolve the higher energy cost of walking. If the authors wish to imply that there is greater efficiency of walking in the group with obesity, then data should be expressed as it is later in the abstract.

In the sentence beginning "Participants..." please include the SD for the % weight loss and also indicate whether the additional average of 5.9% in year 2 represented a statistically significant additional weight loss from year 1.

Introduction:

In comparisons of individuals who have obesity with those who are not overweight or obese (lean), it is important to normalize data to weight or at least be specific as to what is being measured. Specifically, please state if the higher energy cost of walking in individuals is proportional or disproportional to weight.

1st paragraph: The authors are taking Levine et al, 2008, out of context. The study noted that in relatively small populations of individuals with and without obesity, the population with obesity walked significantly less distance both before and after overfeeding. There is no report of actual thermogenesis other than the statement that walking is the major component of NEAT. Overall this paragraph is very confusing because it doesn't separate the expected from the unexpected. For example, in the authors' own work it is expected that absolute NC_w would be higher in proportion to weight. A key finding in their JAP paper (2019) was that "No significant difference was found between groups in relative (per kg of body mass) NetC_w (P 0.13)" or in SMR/kg. This is in contrast to the authors' 2020 paper which did report a higher relative NC_w in individuals with obesity. What is most interesting in this paper is trying to sort out whether differences between study periods are absolute, relative, or in some cases if they just reflect changes in the quality or quantity of physical activity rather than the chemomechanical contractile efficiency of muscle. These distinctions need to be clarified throughout the manuscript.

2nd paragraph: The information in the second paragraph should precede the discussion of the higher net energy cost of walking in individuals with obesity in the first paragraph so that the reader is aware that the authors are discussing a disproportionately increased EE relative to weight. Absolute NC_w is generally not helpful here because of the weight effect.

3rd paragraph: This information is a mechanistic discussion of why relative NC_w might be higher at higher weight and would be more appropriate to the discussion.

4th paragraph: Please delete "therefore" at the start of the last paragraph. There is no physiological reason why loss of weight in and of itself should decrease relative (mass-normalized) NC_w and this statement should be deleted. The proposed changes in muscle efficiency as a consequence of weight loss might be the effector organs for this as hypothesized.

Methods:

Were any participants originally enrolled but then did not have subsequent studies because they did not achieve a 25% weight loss?

Calculations need to be clarified. For example, "W" refers to some number (kcal, joules, etc.) within a certain period of time. Neither is specified. Units should not be expressed as Wxkg⁻¹ but

instead as something like kcal/min/kg to make it more comparable to other data sets. Also, if oxygen consumption is the denominator in W then data should be transformed into kcal or other measures of energy "corrected" for the respiratory exchange ratio. This is standard indirect calorimetry.

Results:

Please report SD whenever presenting results in text. This is particularly true for values expressed as % which are not included in most tables. Please state somewhere whether or not all participants who were prospectively enrolled actually achieved the 25% weight loss or if this is a cohort that was culled from a larger enrolled group.

In participant results please clarify for all post 2 data whether or not the authors are referring to post 2 vs. baseline or post 2 vs. post 1. For example, the text states that there was a significant reduction in FFM only at post 1 but according to table 1 there was a significant reduction in FFM compared to baseline at post 2 and post 1 - just no significant change in FFM from post 1 to post 2.

Please specify the authors' definition of physical activity and what are the units in which it is reported (presumably hours/day but not specified)? Would standing up lecturing or sorting packages be considered physical activity or does one need to be walking? Also, please do not report data as a calculation such as $W \times kg^{-1}$. When presenting data please use the units identified in the text, e.g., $J \times kg^{-1}$. Also, please specify units of time when appropriate. Some variables, such as physical activity, are measured over 24 hours whereas others may be per minute.

Discussion:

The Discussion is well-written.

Studies of weight-reduced individuals suggest that following weight loss there is a significant improvement in muscle efficiency which would translate into an increased W/kg as described in this study. However, they also note a decline in resting energy metabolic rate/kg which should be detected in the SMR. The observation that the increase in relative SMR is not detected until post 2 is not consistent with the suggestion that there is a time in which muscle adaptation to walking (not standing) occurs during which the increased efficiency of muscle may be "masked" by lack of optimal walking style. Specifically, why should mechanical gait changes during the follow-up affect standing metabolic rate?

The implication is that SMR is representative of RMR raises a number of issues. RMR was used by Browning et al and the authors need to clarify the relationship between RMR and SMR in citing this study. The authors cite Vijgen et al, who noted improved mitochondrial function 1 year after bariatric surgery, but should also cite Fernstrom et al (Obes Surg, 2016), who made the

same observation at 6 months.

The authors conclude that a period of adaptation in walking style is necessary to see the improved efficiency of muscle. This is somewhat in contrast to Borges et al (Eur J Appl Physiol, 2018) who found a significant decreased in oxygen consumption/kg on the treadmill that was noted in premenopausal women who were not obese studied before and after weight loss. Also, data should be better contrasted with studies of dietary weight loss (e.g., the reference by Rosenbaum et al) which reported much earlier, but still significant, changes in muscle efficiency after a smaller degree of weight loss. This might reflect the different degrees of weight loss (and therefore greater degree of gait adjustment), different means of weight loss, etc.

END OF COMMENTS

Confidential Review

31-Mar-2021

EDITOR

Dear Dr. Hogan,

Thank you for your correspondence dated 11th of May 2021. We have addressed the criticisms and marked the responses and edits in blue in the reviewed version of the manuscript.

We thank you and the reviewers for their helpful comments, which improved the paper, and we hope that the manuscript is now suitable for publication in Journal of Physiology.

Do not hesitate to contact me for any further queries.

Sincerely,

Davide Malatesta on behalf of the co-authors

Reviewing Editor

We thank the editor for his comments and suggestions. We have considered the remarks of the two reviewers and made amendments when necessary in the revised manuscript. We appreciate your further perusal of the revised manuscript.

We have provided our responses to your comments. Amended sentences are *in italic* with the additional wordings in blue.

[R1] = reviewer (comments).

[A] = authors (responses).

{...} = *text modified in the revised manuscript*.

Number of pages is referred to the corrected word version.

1. The statistical summary document is missing.

The statistical summary document and the raw data files have now been added to the new submission.

2. Results from this study are interesting and adds significantly to the body of literature relevant to adaptive thermogenesis following weight loss. However, concerns raised by the two referees need to be addressed. I encourage you to carefully consider the comments related to the relationship between changes in body weight and changes in walking cost. As suggested, it is also important to normalize the SMR data for the changes in body weight/composition by using an ANCOVA and not by simply dividing W per kg. Also please address the questions of both Referees related to the increase in SMR at point 2. Finally, it is important to appropriately indicate the units of the study outcomes to have a clear and correct message. Thank you.

The concerns raised by the two referees have been addressed in the responses for the reviewers and the manuscript has been modified according to these concerns.

1. The comments related to the relationship between changes in body weight and changes in energy cost of walking have been carefully considered and the Discussion has been modified (please see pp. 23-24, pp. 25-26 and pp. 27-28).
2. Two supplementary linear mixed models with participant set as a random effect and covariate (one with body mass as covariate and one with lean body mass as covariate, respectively) for the absolute SMR. We also used the body mass as covariate for the absolute SMR to be consistent with the same analysis performed and added in the new version of the manuscript for the absolute NC_w with the two updated linear mixed effect analyses of the relationship between conditions and time with body mass or with lean body mass as covariates. In fact, during walking, it is important to take into account the body mass because this latter represents the mass transported directly related to the metabolic cost of walking. Please see “Statistical analysis” (p. 15), Results (pp. 17-18) and Discussion (pp. 23-24).
3. We responded to the questions of both referees related to the increase in SMR at point 2 (please see response #8 for referee #1 and response #17 for referee #2). Moreover, the Discussion has been modified to consider this point (please see pp. 23-24).
4. The calculations used to assess the gross and net metabolic rate or and net energy cost of walking have now been clarified in the Methods of the manuscript (pp. 10-11). We think that, in this new version of the manuscript, the reader can better understand the unit of measurement used in the Results. However, we suggested to reviewer #2 that if it is really important for him to add NC_w in kcal·kg⁻¹·min⁻¹, we can add these values (range or averaged values across all speeds for the 3 groups) in the Results of the manuscript (e.g., in the legend of Figure 1). Please also see responses #2, #12 and #15 for referee #2.

Referee #1

We thank you for this review of our manuscript and for providing comments and suggestions that have helped improve it. We have considered your remarks and made amendments when necessary in the revised manuscript. We appreciate your further perusal of the revised manuscript.

We have provided our responses to your comments. Amended sentences are *in italic* with the additional wordings in blue.

[R1] = reviewer (comments).

[A] = authors (responses).

{...} = *text modified in the revised manuscript*.

Number of pages is referred to the corrected word version.

The authors have examined the effect of substantial weight loss on the energy cost of walking and the mechanical work of walking by comparing these variables in participants with obesity at baseline and then at 2 time points post bariatric surgery. The hypotheses and methods are well-articulated and the work was conducted with appropriate protocols. The analyses are relatively straightforward. My concerns regarding the paper and analyses (and conclusions) are as follows:

We thank the referee for his positive comment.

1. The primary result of the paper is that there appears to be an improvement (decrease) in walking energy cost at post-2, which the authors attribute to increased efficiency. I'm not certain that this is the only interpretation of their data. The changes in walking cost as a % (-25% at post 1, -38% at post 2) match the change in weight (-26% post 1, -32% post 2) almost precisely. This suggests the change in weight is sufficient to explain the change in waking cost, and in fact there is no difference in net walking cost per kg in pre vs post-1. The slightly lower cost/kg at post-2 is completely dependent on the

increased SMR at Post-2. Further, it is *highly* unusual to have an increased SMR (absolute) in subjects who have lost weight (as they do from post1 from post2). Therefore, I strongly encourage the authors to consider the possibility that lower walking cost at post-2 could be an artifact of greater SMR at post-2, which could itself be due to any number of factors: were the subjects truly at a stable SMR at the end of the 5-minute trial? (5 minutes is quite short for measuring resting expenditures). Were the subjects completely fasted for all trials? Were post1 and post2 performed at a similar time of day, with similar physical activity the preceding day? Is there any chance of measurement or machine error in Post 1 or 2? The effect would not have to be large to result in the reported decrease in walking cost at post-2.

We agree with the reviewer that “it is highly unusual to have an increased SMR (absolute) in subjects who have lost weight (as they do from post1 and post2)”. However, there was no significant difference in the absolute SMR between post 1 and post 2 (please see Table 1 and the first sentence of the paragraph “Energetics”, p. 17).

As suggested by the referee, we considered the possibility that reduced NC_w at post 2 could be an artefact of greater SMR at this time point. However, during the experiment, we carefully controlled the confounding factors mentioned by the reviewer and we added this information in the manuscript (please see here below) and we completed our data analysis with body mass and lean body as covariates (please see the Results and Discussion of the manuscript) and the response #8 here below for more details.

According to the questions and confounding factors potentially involved in the assessment of the standing metabolic rate mentioned by the reviewer, we added two new paragraphs in the Methods. Please see the paragraphs “Experimental design” (p. 9) and “Energy cost of walking” (p. 10).

{To standardize the pre-exercise conditions, participants were asked to avoid strenuous exercise the day before each experimental trial, and they reported to the laboratory after a minimum 3-hour fast period and at a similar time of day to avoid circadian variance.}

{Expired gases [oxygen uptake ($\dot{V}O_2$) and CO₂ output ($\dot{V}CO_2$)] were collected (Quark CPET, Cosmed, Italy) breath-by-breath in the standing position for 5 min or more, until the

experimenters visually determined when the stability of $\dot{V}O_2$ and $\dot{V}CO_2$ was reached for each participant. This stability was further visually checked by two blinded and independent investigators who selected the last minute or 1-min time stable window during the data analysis to average $\dot{V}O_2$ and $\dot{V}CO_2$ values and assess the standing metabolic rate during resting (SMR).}

Moreover, the volume and gas calibrations were performed before each trial and, during the period of the experiment, the metabolic card was technically revised each year according to the manufacturer's instructions.

2. If the authors can convincingly rule out that any issue with SMR at post2, the authors should articulate a plausible mechanism for the improved efficiency at Post-2. The authors attribute this improved efficiency to a 'readapted' gait, but they have measured all of the relevant gait parameters and none of them seem to account for the improved ratio of metabolic energy / mechanical work at post-2. Indeed, recovery% drops at post1 and post2. Do the authors propose that the muscles themselves become more efficient? If not, which aspects of gait could be changing in a favorable manner to improve efficiency that aren't already captured and analyzed? Of course, if the energy measurements at post-2 are low because of an artifact in the measurement of SMR at post-2, then the apparent improvement in efficiency at post-2 will likely fall away, making this point moot (and substantially changing the framing of the Discussion).

In our response #8 here below and in the new version of the Discussion (pp. 23-24), we explained the reasons why we can rule out any issue with SMR at post 2.

We then modified the Discussion to better clarify and articulate a plausible mechanism for the improved efficiency at post 2 as recommended by the referee (please see pp. 24-28). Briefly, the biomechanical changes almost obtained in the first part of the follow-up (from baseline to post-1) result in a reorganisation of the gait pattern into a gait more similar to the normal body mass loss. The less knee flexed lower limbs should have induced a decreased NCw, already at post 1, as previously suggested by Grabowski et al. (2005), Ortega & Farley (2005) and Griffin et al. (2003). However, this expected enhanced walking economy was not obtained at post 1 because the increased Wext associated with a reduced

Recovery at this time point compared to baseline (reversing the more skilful recovery developed in individuals with obesity vs normal body mass individuals to minimize the increase in NCw in the former) may penalize NCw and mask the likely expected improvement in gait economy at muscle level (pp. 25-26). In the second part of the follow-up (from post 1 to post 2), the “mechanical stability” of the gait pattern would be a period of adaptation to the new gait pattern needed to an improvement in mechanical efficiency obtained by an enhancement in muscle contraction efficiency and an increase in propulsive efficiency (i.e., the transformation of the positive work performed by muscle to mechanical work) [the two components of overall efficiency of walking (Cavagna, 2017)] (p. 26 and pp. 27-28).

{The less knee flexed lower limbs at post 1 than at baseline may induce advantageous joint moments associated with lower muscle activation (Grabowski et al., 2005; Ortega & Farley, 2005) and NCw (Griffin et al., 2003). However, the increased W_{ext} , which accounts for about one-half of NC_w in adults (Grabowski et al., 2005), at post 1 compared to baseline may contribute to blunt this expected improvement in walking economy maintaining similar NCw between these two-time points.}

{Corroborating this hypothesis, our results showed a significant increase in mechanical efficiency only at post 2, due to the similar mass-normalized W_{tot} and decrease in relative NCw at this time point compared with those at post 1. This improved mechanical efficiency may be due to an enhancement in muscle contraction efficiency but also an increase in propulsive efficiency (i.e., the transformation of the positive work performed by muscle to mechanical work) [the two components of overall efficiency of walking (Cavagna, 2017)].}

{Moreover, the “mechanical stability” of the gait pattern, between post 1 and post 2, would be a period of adjustment to the new gait pattern, developed from baseline to post 1 and essentially due to the very large body mass loss, necessary to an enhancement in propulsive efficiency. This improvement, obtained during walking with the new gait pattern in the second part of the follow-up (i.e., adaptive and behavioural changes), may be due to lower muscle activations and by making the muscles work in more favourable conditions (e.g., decreased volume of active muscle operating at advantageous length and/or velocities). This

also confirms that human walking optimization is a compromise between saving energy via pendulum-like mechanism and making muscles work efficiently (Massaad et al., 2007).

The enhanced mechanical efficiency at post 2 corroborates previous findings showing an improved muscle metabolic economy and mitochondrial function after diet- (Newcomer et al., 2001) or bariatric surgery-induced (Nijhawan et al., 2013; Vijgen et al., 2013; Fernstrom et al., 2016) body mass loss. Neuroendocrine changes (e.g., a decrease in triiodothyronine concentration, an increase in sympathetic tone and a reduction in parasympathetic tone) might enhance skeletal muscle contractile energy efficiency after body mass loss (Rosenbaum & Leibel, 2010; Galgani & Santos, 2016). However, some studies reported that the cycling efficiency (Rosenbaum et al., 2003) and NC_w (Borges et al., 2018) were improved directly after shorter body mass loss interventions and, thus, somewhat in contrast with our delayed enhancement in NC_w and mechanical efficiency only at post 2. This might reflect the different degree of body mass loss obtained in individuals with overweight by these previous studies (-10-12 kg corresponding to 10-16% of the initial body mass) compared with that of the present study at post 1 (-30 kg or -25% of the body mass at baseline). Moreover, this difference indirectly confirms that a very large body mass loss, inducing a greater degree of gait pattern changes, would need a period of adaptation to the new gait pattern to improve its mechanical efficiency.}

3. Introduction, paragraph 1: It's true that weight gain must be due to energy imbalance, but the discussion here implies that weight gain and obesity result from lower energy expenditure. The available evidence points to overconsumption of calories as a much larger factor in the development of obesity than a decrease in expenditure. See, for example, work by Prentice, Dugas, Luke, or Pontzer showing that total daily energy expenditures are not lower in people with obesity and that total daily expenditures do not correspond with daily physical activity.

According to the comments of both reviewers, the paragraphs #1 and #2 of the Introduction were modified. Please see pp. 4-5.

{Obesity has been recognized as a significant public health issue worldwide, with a prevalence that has been continuously increasing over the past decades, leading to a variety of chronic diseases and increasing health care costs (Collaboration, 2016). The causes of obesity are multifactorial, but the energy imbalance (i.e., lower daily energy expenditure compared to daily energy intake) seems to be the main driver leading to weight gain (Yoo, 2018). Although a reduction in physical activity level (Guthold et al., 2018) may be involved in a decreased daily energy expenditure, the role of physical activity in energy balance (Blundell et al., 2015) and weight loss remains controversial (Pontzer et al., 2016). However, independently of weight loss, physical activity is crucially important for improving overall health and fitness in the prevention and treatment of obesity (Luke & Cooper, 2013).

Walking is the most common modality to promote daily physical activity, reduce sedentary time and improve health (Murtagh et al., 2015). However, the higher net energy cost of walking (NC_w : the energy expended above the resting energy expenditure per unit distance) in individuals with obesity than in their lean counterparts may lead to motor impairments and fatigue and contribute to an increase in physical inactivity and sedentary time in daily life in the former (Levine et al., 2005; Levine et al., 2008). This may thus reduce the efficacy of walking in weight management programmes. Moreover, the absolute NC_w ($J \cdot m^{-1}$) but also the relative NC_w (i.e., normalized by body mass; $J \cdot kg^{-1} \cdot m^{-1}$) are higher in individuals with obesity than in lean adults (Browning et al., 2006; Browning & Kram, 2007; Fernandez Menendez et al., 2020), suggesting that body mass is the main but not the only factor affecting the lower walking economy in this population. Therefore, understanding the mechanisms (i.e., mass driven, gait pattern and behavioural changes) involved in this extra cost of walking in adults with obesity is pivotal to improving gait economy and optimizing the use of walking to promote daily physical activity and improve health in these individuals.}

4. Introduction: Defining the net cost of walking here, may want to include that this is the energy expended over and above resting energy expenditure.

As suggested by the referee the definition of the net energy cost of walking is completed. Please see the Introduction (first paragraph, p. 4).

{However, the higher net energy cost of walking (NC_w : the energy expended above the resting energy expenditure per unit distance) in individuals with obesity than in their lean counterparts may lead to motor impairments and fatigue and contribute to an increase in physical inactivity and sedentary time in daily life in the former (Levine et al., 2005; Levine et al., 2008).}

5. Introduction: Regarding the greater cost of walking with obesity and its effects on NEAT: it seems equally plausible that the greater walking costs (J / km) would result in greater NEAT with obesity, even if steps/day is reduced.

Levine et al. (Science 2005) have reported that “Our analysis revealed that obese participants were seated for 164 min longer per day than were lean participants (Fig. 1A). Correspondingly, lean participants were upright for 152 min longer per day than obese participants.”. This was associated with a decreased relative NEAT (kcal/kg of body mass) in individuals with obesity than in lean counterparts. This reference has now been added in the new version of the Introduction. Moreover, as already stated in the responses #3, the first 2 paragraphs of the Introduction have now been modified according to the comments of the two referees.

6. Methods: Please briefly review the protocol for determining preferred walking speed.

The protocol for determining preferred walking speed (PWS) has now been added in the Methods (paragraph “Preferred walking speed”, p. 10).

{Briefly, participants started to walk on the treadmill at the lowest experimental speed ($0.56 \text{ m}\cdot\text{s}^{-1}$) without receiving any feedback regarding their speed. The speed was gradually increased until the participant subjectively identified the PWS and maintained for 1 min and slightly modified according to the participant’s instructions. This procedure was repeated starting from the highest experimental speed ($1.67 \text{ m}\cdot\text{s}^{-1}$) or from the previously assessed PWS + $0.42 \text{ m}\cdot\text{s}^{-1}$ (when PWS > $1.25 \text{ m}\cdot\text{s}^{-1}$) and then gradually reducing the speed to the

individual subjective PWS. The average of the two speeds selected by each participant (i.e., increasing and decreasing speed trials) was considered the final PWS.}

7. See comment 1 above: please provide more information on time of day, whether subjects were fasted, how SMR was established to be stable, etc.

As suggested by the referee, we added two new paragraphs in the Methods. Please see the paragraphs “Experimental design” (pp. 8-9) and “Energy cost of walking” (p. 10).

{To standardize the pre-exercise conditions, participants were asked to avoid strenuous exercise the day before each experimental trial, and they reported to the laboratory after a minimum 3-hour fast period and at a similar time of day to avoid circadian variance.}

{Expired gases [oxygen uptake ($\dot{V}O_2$) and CO₂ output ($\dot{V}CO_2$)] were collected (Quark CPET, Cosmed, Italy) breath-by-breath in the standing position for 5 min or more, until the experimenters visually determined when the stability of $\dot{V}O_2$ and $\dot{V}CO_2$ was reached for each participant. This stability was further visually checked by two blinded and independent investigators who selected the last minute or 1-min time stable window during the data analysis to average $\dot{V}O_2$ and $\dot{V}CO_2$ values and assess the standing metabolic rate during resting (SMR).}

8. Results: In reporting the SMR pre- and post-weight loss, please be aware that it's inappropriate to compare the simple ratios of energy/kg. Using this simple ratio assumes implicitly that the relationship between SMR and body mass (or lean mass) is a straight line with an intercept at y=0. That is very rarely the case with resting or total daily metabolic rates. The appropriate way to examine the relationship with body size is to use regression analysis or ANCOVA, with lean mass as the covariate and time point (pre, post1, post2) as a categorical variable. (you could add other variables to lean mass, but with a small sample size that's probably not advisable). [note: this may also affect the discussion of SMR changes in the Discussion page 21]. See: D. B. Allison, F. Paultre,

M. I. Goran, E. T. Poehlman, S. B. Heymsfield, Statistical considerations regarding the use of ratios to adjust data. Int J Obes Relat Metab Disord 19, 644-652 (1995).

As suggested by the reviewer, we added in the manuscript two supplementary linear mixed models with participant set as a random effect and covariate (one with body mass as covariate and one with lean body mass as covariate, respectively) for the absolute SMR. We also used the body mass as covariate for the absolute SMR to be consistent with the same analysis performed and added in the new version of the manuscript for the absolute NCw with the two updated linear mixed effect analyses of the relationship between conditions and time with body mass or with lean body mass as covariates. In fact, during walking it is important to take into account the body mass because this latter represents the mass transported directly related to metabolic cost of walking.

Please see the paragraph “Statistical analysis” (p. 15), Results (pp. 17-18 and Table 1).

{An updated linear mixed model with body mass as covariate and one with lean body mass as covariate were used for the absolute SMR. }

{An updated linear mixed effects analysis of the relationships between conditions and time with body mass as a covariate and one with lean body mass were used for the absolute NCw.}

{The mixed linear model with body mass as covariate ($P = 0.042$) indicated that there was no significant difference in the absolute SMR during follow-up ($P = 0.13$; Table 1).}

{The mixed linear model with lean body mass as covariate ($P = 0.11$) showed that the absolute SMR was significantly decreased at post 1 compared to the baseline ($P = 0.02$) with no difference between baseline and post 2 ($P = 0.14$) and between the two follow-ups ($P = 0.95$; Table 1).}

{The mixed linear model for the absolute NC_w with body mass as covariate ($P = 0.003$) confirmed these results with a significantly decreased absolute NC_w at post 2 compared to

that at baseline (mean difference: -19%; $P = 0.045$) and post 1 (mean difference: -11%; $P < 0.0001$) while there was no significant difference between baseline and post 1 ($P = 0.49$).

{These results have been confirmed by the mixed linear mixed model for the absolute NC_w with lean body mass as covariate indicating that the absolute NC_w significantly decrease between baseline and post 1 (mean difference: -21%; $P < 0.0001$) and post 2 (mean difference: -35%; $P < 0.0001$) and between the two follow-ups (mean difference: -17%; $P < 0.0001$ for both).}

We also modified the Discussion (pp. 23-24) and we introduced and discussed the controversial about the simple-ratio and regression or ANCOVA analysis in the comparison of resting metabolic rate before and after body mass loss. However, the results of the mixed linear model for the absolute NC_w with body mass (or lean body mass) as covariate confirmed those of the mass-normalized NC_w analysis (ratio analysis). This demonstrates that, for the main outcome of the study (the relative NC_w), the ratio analysis can be correctly used to analyse our data as previously suggested by Packard and Boardman (Comparative Biochemistry and Physiology a-Molecular and Integrative Physiology 1999). Therefore, in our opinion, these findings clearly demonstrate that reduced NC_w at post 2 is not an artefact of greater SMR at this time point but it is related to an improved mechanical efficiency (muscle contraction efficiency and propulsive efficiency) at Post 2.

{Although an increase in the relative SMR after a nonsurgical intervention (Peyrot et al., 2010) and in relative resting metabolic rate after bariatric surgery (de Cleva et al., 2018) has already been reported, the comparison of these mass-normalized measures of standing or resting metabolic rate (simple ratio-based assessments) before and after body mass loss is questioned (Cooper & Berman, 1994; Packard & Boardman, 1999; Browning et al., 2018). In fact, using body mass as covariate, we did not find any differences in the absolute SMR during follow-up, attesting that ratio might introduce problems with respect to statistical analysis and interpretation of SMR data (Browning et al., 2018). However, the results of the mixed linear model for the absolute NC_w with body mass as covariate confirmed those of the mass-normalized NC_w analysis (ratio analysis). This demonstrates that, for the main outcome of the study (the relative NC_w), i) the ratio analysis can be correctly used to analyse our data

(Packard & Boardman, 1999) and ii) the decrease in body mass is not the only factor involved in the reduced relative NC_w after bariatric surgery.}

Referee #2

We thank you for this review of our manuscript and for providing comments and suggestions that have helped improve it. We have considered your remarks and made amendments when necessary in the revised manuscript. We appreciate your further perusal of the revised manuscript.

We have provided our responses to your comments. Amended sentences are *in italic* with the additional wordings in **blue**.

[R1] = reviewer (comments).

[A] = **authors (responses)**.

{...} = *text modified in the revised manuscript*.

Number of pages is referred to the corrected word version.

General Comments

1. This is a study of changes in energy expenditure by muscle during walking before and after bariatric surgery. The authors' main finding is that increased muscle efficiency, defined as work/kg, is not significantly increased until 1 year after bariatric surgery. This is very interesting. The pre- and post- bariatric surgery studies and overall attention to detail are strengths, but the manuscript is difficult to follow.

We thank the reviewer for his positive comment about our study.

The increased mechanical efficiency (%), indirectly attesting for an increased muscle efficiency (please see the Discussion of the manuscript), is defined as the ratio between the total mechanical work (W_{tot} ; $J \cdot kg^{-1} \cdot m^{-1}$) and net metabolic/energy cost of walking (NC_w ; $J \cdot kg^{-1} \cdot m^{-1}$) multiplied by 100 and its unit of measurement is the percentage and not “work/kg”.

The manuscript has been modified according to the referees' comments and we hope that this new version of the manuscript is easier to follow (please see our responses here below and the new version of the manuscript).

2. Energy expenditure is not directly expressed since data are generally presented with "work performed" rather than energy burned during exercise. Energy expenditure and work are interchanged in a very confusing manner. For example, W_{tot} is defined as total mechanical work, but standing metabolic rate, which should really be somewhat equivalent of a standing RMR, is also defined as work. Since no object is really moving during SMR - work seems to be an inappropriate measure. This terminology is used in other papers by this group but can be difficult to follow. Clarification would be helpful.

We thank the referee for this comment that allows us to clarify the calculations used to assess energy expenditure during standing and walking in response #12 and in the Methods of the manuscript (please see pp. 10-11 and here below).

The standing metabolic rate is not presented as "work performed" [in J and normalized to the walking distance ($J \cdot m^{-1}$ or $J \cdot kg^{-1} \cdot m^{-1}$), as used for the mechanical works] throughout the manuscript, but as "metabolic power" or "metabolic rate" in W or in $W \cdot kg^{-1}$ ($J \cdot s^{-1}$ or $J \cdot J \cdot kg^{-1} \cdot s^{-1}$) (please see Table 1).

The average $\dot{V}O_2$ values ($mlO_2 \cdot kg^{-1} \cdot min^{-1}$) during standing and walking were converted in $W \cdot kg^{-1}$ ($= J \cdot kg^{-1} \cdot s^{-1}$) using the energy equivalent of 1 L of O_2 [$J \cdot (1 L O_2)^{-1}$] (see equation here below) calculated using the average RER values over the same time (Astrand, 1986) [metabolic rate/power ($W \cdot kg^{-1}$) = ($LO_2 \cdot kg^{-1} \cdot s^{-1}$) \cdot [$J \cdot (1 L O_2)^{-1}$] = $J \cdot kg^{-1} \cdot s^{-1}$ = $W \cdot kg^{-1}$].

$$\text{Energy equivalent of 1 L of } O_2 \text{ (kJ)} = 21.13 \left[\frac{(RER-0.7)}{0.3} \right] + 19.6 \left[\frac{(1-RER)}{0.3} \right]$$

During walking, the net metabolic power ($W \cdot kg^{-1} = J \cdot kg^{-1} \cdot s^{-1}$) is then divided by walking speed ($m \cdot s^{-1}$) to obtain the metabolic/energy cost of walking ($J \cdot kg^{-1} \cdot m^{-1}$) that represents the energy expenditure above resting per unit of distance (i.e., assessment of walking economy).

We prefer to use "W" as the unit of measurement of power (i.e., metabolic rate/power) and "J" as the unit of measurement of energy (metabolic/energy cost of walking; $J \cdot kg^{-1} \cdot m^{-1}$) and work (mechanical work performed per unit of distance and normalized by body mass; $J \cdot kg^{-1} \cdot m^{-1}$) because "W" and "J" are the units of measurement used in the international system of units for these quantities (power and energy/work, respectively). Moreover, these units

are usually adopted and presented in seminal manuscripts about energetics and mechanics of locomotion (e.g., di Prampero, Int J Sports Med 1986) and in the specific literature about gait energetics and mechanics in individuals with obesity (please see the manuscripts about this topic of Browning et al. and our own works). This makes easier a specific comparison between our values and the previous ones.

Besides, to calculate mechanical efficiency ($W_{\text{tot}} \cdot NC_w^{-1}$), W_{tot} ($\text{J} \cdot \text{kg}^{-1} \cdot \text{m}^{-1}$) and NC_w^{-1} ($\text{J} \cdot \text{kg}^{-1} \cdot \text{m}^{-1}$) must be expressed using the same units of measurement. In fact, the mechanical efficiency assesses the fraction of the amount of metabolic energy (i.e., metabolic input) that can be transformed into mechanical work (i.e., mechanical output) (please see for review Peyré-Tartaruga & Coertjens, Frontiers in Physiology 2018).

Abstract

3. Throughout the manuscript please use "people first" language. For example, please use the term "individuals with obesity" rather than "obese individuals".

The manuscript has now been revised for using “people first” language.

4. The first sentence is self-contradictory. Specifically, improving the higher energy cost of walking cannot result in lower energy expenditure and it is unlikely that a large loss of mass would somehow resolve the higher energy cost of walking. If the authors wish to imply that there is greater efficiency of walking in the group with obesity, then data should be expressed as it is later in the abstract.

According to the suggestion of the referee, the first sentence of the Abstract and third point of the “Key points summary” have now been changed (pp. 2-3).

{Understanding the mechanisms involved in the higher energy cost of walking (NC_w : the energy expenditure above resting per unit distance) in adults with obesity is pivotal to optimizing the use of walking in weight management programmes.}

{Understanding the mechanisms (i.e., mass driven, gait pattern and behavioural changes) involved in this extra cost of walking in adults with obesity is pivotal to optimizing the use of walking to promote daily physical activity and improve health in these individuals.}

5. In the sentence beginning "Participants..." please include the SD for the % weight loss and also indicate whether the additional average of 5.9% in year 2 represented a statistically significant additional weight loss from year 1.

The SD for weight loss and the p-value of the significant difference between post 1 and post 2 have now been added to the Abstract (please see p. 3).

In the new version of the manuscript, there are some slight changes in the numeric % values (e.g., for this result: 5.9%, in the first version, vs 6.1%, in this new version). This was due to the fact that, in the first version of the manuscript, we reported the percentage of the differences among the average values of the experimental variables at the 3-time points for the group to present a general indication of the % time change for the reader. For this reason, we did not add the SD. The SDs of these values expressed as % have now been calculated for each participant and then averaged for the group and added in the new version of the manuscript (Abstract and Results).

{Participants lost $25.7 \pm 3.4\%$ of their body mass at post 1 (6.6 months; $P < 0.001$) and $6.1 \pm 4.9\%$ more at post 2 (12 months; $P = 0.014$).}

Introduction

6. In comparisons of individuals who have obesity with those who are not overweight or obese (lean), it is important to normalize data to weight or at least be specific as to what is being measured. Specifically, please state if the higher energy cost of walking in individuals is proportional or disproportional to weight.

According to the comments of both reviewers, the paragraphs #1 and #2 of the Introduction were modified (please see my responses #7 and #8 here below) and the information about

the proportional and disproportional relationship between the energy cost of walking and body mass has now been clearly stated in the Introduction (please see p. 4).

{This may thus reduce the efficacy of walking in weight management programmes. Moreover, the absolute NCw ($J\cdot m^{-1}$) but also the relative NCw (i.e., normalized by body mass; $J\cdot kg^{-1}\cdot m^{-1}$) are higher in individuals with obesity than in lean adults (Browning et al., 2006; Browning & Kram, 2007; Fernandez Menendez et al., 2020), suggesting that body mass is the main but not the only factor affecting the lower walking economy in this population.}

7. 1st paragraph: The authors are taking Levine et al, 2008, out of context. The study noted that in relatively small populations of individuals with and without obesity, the population with obesity walked significantly less distance both before and after overfeeding. There is no report of actual thermogenesis other than the statement that walking is the major component of NEAT. Overall this paragraph is very confusing because it doesn't separate the expected from the unexpected. For example, in the authors' own work it is expected that absolute NC-w would be higher in proportion to weight. A key finding in their JAP paper (2019) was that "No significant difference was found between groups in relative (per kg of body mass) NetCw (P 0.13)" or in SMR/kg. This is in contrast to the authors' 2020 paper which did report a higher relative NCw in individuals with obesity. What is most interesting in this paper is trying to sort out whether differences between study periods are absolute, relative, or in some cases if they just reflect changes in the quality or quantity of physical activity rather than the chemomechanical contractile efficiency of muscle. These distinctions need to be clarified throughout the manuscript.

According to the suggestions of the referee, the first and the second paragraph has now been modified (please pp. 4-5).

{Obesity has been recognized as a significant public health issue worldwide, with a prevalence that has been continuously increasing over the past decades, leading to a variety of chronic diseases and increasing health care costs (Collaboration, 2016). The causes of obesity are multifactorial, but the energy imbalance (i.e., lower daily energy expenditure

compared to daily energy intake) seems to be the main driver leading to weight gain (Yoo, 2018). Although a reduction in physical activity level (Guthold et al., 2018) may be involved in a decreased daily energy expenditure, the role of physical activity in energy balance (Blundell et al., 2015) and weight loss remains controversial (Pontzer et al., 2016). However, independently of weight loss, physical activity is crucially important for improving overall health and fitness in the prevention and treatment of obesity (Luke & Cooper, 2013).

Walking is the most common modality to promote daily physical activity, reduce sedentary time and improve health (Murtagh et al., 2015). However, the higher net energy cost of walking (NC_w: the energy expended above the resting energy expenditure per unit distance) in individuals with obesity than in their lean counterparts may lead to motor impairments and fatigue and contribute to an increase in physical inactivity and sedentary time in daily life in the former (Levine et al., 2005; Levine et al., 2008). This may thus reduce the efficacy of walking in weight management programmes. Moreover, the absolute NC_w (J·m⁻¹) but also the relative NC_w (i.e., normalized by body mass; J·kg⁻¹·m⁻¹) are higher in individuals with obesity than in lean adults (Browning et al., 2006; Browning & Kram, 2007; Fernandez Menendez et al., 2020), suggesting that body mass is the main but not the only factor affecting the lower walking economy in this population. Therefore, understanding the mechanisms (i.e., mass driven, gait pattern and behavioural changes) involved in this extra cost of walking in adults with obesity is pivotal to improving gait economy and optimizing the use of walking to promote daily physical activity and improve health in these individuals.}.

8. 2nd paragraph: The information in the second paragraph should precede the discussion of the higher net energy cost of walking in individuals with obesity in the first paragraph so that the reader is aware that the authors are discussing a disproportionately increased EE relative to weight. Absolute NC_w is generally not helpful here because of the weight effect.

The third paragraph of the Introduction (2nd paragraph of the first version of the manuscript) has now been preceded by a paragraph in which the disproportional increased NC_w relative to body mass is clearly stated (2nd paragraph of the new version; pp. 4-5) and it

is now focused only on relative NCw (per kg of body mass – mass-normalized). Please see p. 5.

{Recently, Fernández Menéndez et al. (Fernandez Menendez et al., 2020) showed that the relative NCw was 19% higher in adults with class III obesity than in their lean counterparts and was associated with a lower amount of mass-normalized external mechanical work [i.e., work performed to lift and accelerate the centre of mass (CM) relative to the surroundings, W_{ext}], higher pendular recovery (i.e., gait energy saving mechanism due to the exchange between potential and kinetic energy of CM that minimizes W_{ext}) but similar mass-normalized internal (i.e., work required to move the limbs with respect to CM, W_{int}) and total (i.e., $W_{tot} = W_{ext} + W_{int}$) mechanical works. As a consequence, the mechanical efficiency ($W_{tot} \cdot NC_w^{-1}$) was reduced in individuals with obesity due to their higher relative NC_w , likely related to muscle level differences (e.g., more muscle fibre work or force and/or poorer muscle efficiency in individuals with obesity than in lean adults). This may be associated with the more erect gait pattern (i.e., reduced hip and knee flexion and increased ankle plantar flexion) (DeVita & Hortobagyi, 2003; Fernandez Menendez et al., 2018), which requires larger muscle activation and makes muscles operate in disadvantageous lengths and/or velocities, thereby inducing poorer muscle efficiency (Massaad et al., 2007) in individuals with obesity than in lean individuals.}

9. 3rd paragraph: This information is a mechanistic discussion of why relative NCw might be higher at higher weight and would be more appropriate to the discussion.

We understand the point of view of the reviewer. However, we think that is paragraph is really important and pivotal in the Introduction because we can introduce 1) the effect of bariatric surgery (i.e., massive body mass loss) on the gait pattern (i.e., “mechanical plasticity” and “gait pattern reversibility” after a weight loss) (Hortobagyi et al., J Appl Physiol 2011) and 2) the relationship between the decrease in NC_w and improved muscle efficiency after nonsurgical intervention previously shown by Peyrot et al. (Med Sci Sports Exerc 2010). These important points, completed by our mechanical and energetic analysis in adults with obesity, allow the reader to understand the relevance and originality of our study. For this reason, we decided to maintain this important paragraph in the Introduction.

10. 4th paragraph: Please delete "therefore" at the start of the last paragraph. There is no physiological reason why loss of weight in and of itself should decrease relative (mass-normalized) NCw and this statement should be deleted. The proposed changes in muscle efficiency as a consequence of weight loss might be the effector organs for this as hypothesized.

As suggested by the reviewer the term "therefore" has now been deleted. Please see p. 6.

{The purpose of this study was to investigate the mechanics, energetics and mechanical efficiency of walking after a very large body mass loss induced by bariatric surgery in adults with obesity (1-year follow-up).}

Methods

11. Were any participants originally enrolled but then did not have subsequent studies because they did not achieve a 25% weight loss?

As reported in the first version of the manuscript (Discussion, p. 28): "Only 11 individuals agreed to participate in the protocol, and 2 of them dropped out after the baseline assessments for reasons independent of the study's protocol.". There were no other dropouts in the study protocol.

12. Calculations need to be clarified. For example, "W" refers to some number (kcal, joules, etc.,) within a certain period of time. Neither is specified. Units should not be expressed as $W \times kg^{-1}$ but instead as something like kcal/min/kg to make it more comparable to other data sets. Also, if oxygen consumption is the denominator in W then data should be transformed into kcal or other measures of energy "corrected" for the respiratory exchange ratio. This is standard indirect calorimetry.

We thank the referee for his comment that allows us to clarify the calculations used to assess the gross and net metabolic rate or power (i.e., the energy expenditure per unit of time) and net energy/metabolic cost of walking (i.e., the energy expenditure per distance

travelled) (di Prampero, Int J Sports Med 1986) in the Methods of the manuscript (please see p. 10 and here below).

The average $\dot{V}O_2$ values ($\text{mlO}_2 \cdot \text{kg}^{-1} \cdot \text{min}^{-1}$) during standing and walking were converted in $\text{W} \cdot \text{kg}^{-1}$ ($= \text{J} \cdot \text{kg}^{-1} \cdot \text{s}^{-1}$) using the energy equivalent of 1 L of O_2 [$\text{J} \cdot (\text{1 L O}_2)^{-1}$] (see equation here below) calculated using the average RER values over the same time (Astrand, 1986) [$\text{metabolic rate/power } (\text{W} \cdot \text{kg}^{-1}) = (\text{LO}_2 \cdot \text{kg}^{-1} \cdot \text{s}^{-1}) \cdot [\text{J} \cdot (\text{1 L O}_2)^{-1}] = \text{J} \cdot \text{kg}^{-1} \cdot \text{s}^{-1} = \text{W} \cdot \text{kg}^{-1}$].

$$\text{Energy equivalent of 1 L of O}_2 \text{ (kJ)} = 21.13 \left[\frac{(\text{RER}-0.7)}{0.3} \right] + 19.6 \left[\frac{(1-\text{RER})}{0.3} \right]$$

During walking, the net metabolic power ($\text{W} \cdot \text{kg}^{-1} = \text{J} \cdot \text{kg}^{-1} \cdot \text{s}^{-1}$) is then divided by walking speed ($\text{m} \cdot \text{s}^{-1}$) to obtain the metabolic/energy cost of walking ($\text{J} \cdot \text{kg}^{-1} \cdot \text{m}^{-1}$) that represents the exergy expenditure above resting per unit of distance (i.e., assessment of walking economy).

We prefer to use “W” as the unit of measurement of power (i.e., metabolic rate/power) and “J” as the unit of measurement of energy (metabolic/energy cost of walking; $\text{J} \cdot \text{kg}^{-1} \cdot \text{m}^{-1}$) and work (mechanical work performed per unit of distance and normalized by body mass; $\text{J} \cdot \text{kg}^{-1} \cdot \text{m}^{-1}$) because “W” and “J” are the units of measurement used in the international system of units for these quantities (power and energy/work, respectively). Moreover, these units are usually adopted and presented in seminal manuscripts about energetics and mechanics of locomotion (e.g., di Prampero, Int J Sports Med 1986) and in the specific literature about gait energetics and mechanics in individuals with obesity (please see the manuscripts about this topic of Browning et al. and our own works). This makes easier a specific comparison between our values and the previous ones.

Besides, to calculate mechanical efficiency ($W_{\text{tot}} \cdot \text{NC}_w^{-1}$), W_{tot} ($\text{J} \cdot \text{kg}^{-1} \cdot \text{m}^{-1}$) and NC_w^{-1} ($\text{J} \cdot \text{kg}^{-1} \cdot \text{m}^{-1}$) must be expressed using the same units of measurement. In fact, the mechanical efficiency assesses the fraction of the amount of metabolic energy (i.e., metabolic input) that can be transformed into mechanical work (i.e., mechanical input) (please see for review Peyré-Tartaruga & Coertjens, Frontiers in Physiology 2018).

However, if the reviewer thinks that is really important to add in the manuscript some information about NC_w in $\text{kcal} \cdot \text{kg}^{-1} \cdot \text{min}^{-1}$, we can add these values (range or averaged values

across all speeds for the 3 groups) in the Results of the manuscript (e.g., in the legend of Figure 1).

{ $\dot{V}O_2$ values ($mLO_2 \cdot kg^{-1} \cdot min^{-1}$) from the last minute (i.e., steady state) were averaged. With the average RER over the same last minute, the energy equivalent of 1 L of O_2 was determined [$J \cdot (1 L O_2)^{-1}$] (Astrand, 1986) and then multiplied by the average $\dot{V}O_2$ ($L O_2 \cdot kg^{-1} \cdot s^{-1}$) at steady state during the last minute of walking for each speed to calculate the gross metabolic rate in $W \cdot kg^{-1}$. The same procedure was applied to convert the average $\dot{V}O_2$ during standing in $W \cdot kg^{-1}$. The net metabolic rate ($W \cdot kg^{-1}$) was calculated by subtracting the SMR from the gross metabolic rate and then divided by the corresponding walking speed ($m \cdot s^{-1}$) to determine NC_w . This latter was expressed in absolute ($J \cdot m^{-1}$) and relative [i.e., normalized by the body mass ($J \cdot kg^{-1} \cdot m^{-1}$) and lean body mass ($J \cdot kgLM^{-1} \cdot m^{-1}$)] values throughout this manuscript.}

Results

13. Please report SD whenever presenting results in text. This is particularly true for values expressed as % which are not included in most tables. Please state somewhere whether or not all participants who were prospectively enrolled actually achieved the 25% weight loss or if this is a cohort that was culled from a larger enrolled group.

In the first version of the manuscript, we reported the percentage of the differences among the average values of the experimental variables at the 3-time points for the group to present a general indication of the % time change for the reader. For this reason, we did not add the SD. The SDs of these values expressed as % have now been calculated for each participant and then averaged for the group and added in the new version of the manuscript.

Only one participant decreased his body mass less than the 25% expected at post 1 obtaining -20% at this time point. Other two participants are ~2.5 kg (2-2.5%) less than 25% expected. For these two participants, these minor differences between the actual and expected body mass loss were due because the experimenters regularly checked by phone the body mass of the participants and then when the 25% was reached the participants are asked to come to the laboratory where the body mass was actually assessed. A slight

difference between the body mass assessed at home by the participants and that measured at the laboratory can explain this gap. Please see Results (p. 16).

{Only one participant decreased his body mass less than the 25% expected at post 1 obtaining -20% at this time point.}

14. In participant results please clarify for all post 2 data whether or not the authors are referring to post 2 vs. baseline or post 2 vs. post 1. For example, the text states that there was a significant reduction in FFM only at post 1 but according to table 1 there was a significant reduction in FFM compared to baseline at post 2 and post 1 - just no signification change in FFM from post 1 to post 2.

The text of the Results of the manuscript has now been rewritten according to the suggestion of the reviewer. Please see pp. 16-20.

15. Please specify the authors' definition of physical activity and what are the units in which it is reported (presumably hours/day but not specified)? Would standing up lecturing or sorting packages be considered physical activity or does one need to be walking? Also, please do not report data as a calculation such as $W \times kg^{-1}$. When presenting data please use the units identified in the text, e.g., $J \times kg^{-1}$. Also, please specify units of time when appropriate. Some variables, such as physical activity, are measured over 24 hours whereas others may be per minute.

The physical activity level was assessed using a self-reported questionnaire (Baecke et al., Am J Clin Nutr 1982) evaluating physical activity at work and physical activity during leisure, excluding sports and sports during leisure time over 6 months before the baseline, between the baseline and post 1 and post 1 and post 2. Different scores were used to quantify work, leisure and sport activities, altogether resulting in a total physical activity score. This information has now been added in the Methods of the manuscript (p. 9).

{Physical activity level. Each participant completed a self-reported measurement of a habitual physical activity questionnaire assessing physical activity at work and physical

activity during leisure, excluding sports and sports during leisure time over 6 months before the baseline, between the baseline and post 1 and post 1 and post 2. Different scores were used to quantify work, leisure and sport activities, altogether resulting in a total physical activity score (Baecke et al., 1982).}

The calculations used to assess the gross and net metabolic rate or power (di Prampero, Int J Sports Med 1986) and net energy cost of walking have now been clarified in the Methods of the manuscript (please see p. 10 and responses #2 and #12 here above). We think that, in this new version of the manuscript, the reader can better understand the unit of measurement used in the Results. However, if the reviewer thinks that is really important to add some other information or clarifications about for example NC_w in $\text{kcal}\cdot\text{kg}^{-1}\cdot\text{min}^{-1}$, as previously suggested (response #12 here above), we can add these values (range or averaged values across all speeds for the 3 groups) or other information in the Results of the manuscript (e.g., for NC_w in $\text{kcal}\cdot\text{kg}^{-1}\cdot\text{min}^{-1}$ in the legend of Figure 1).

Discussion:

16. The Discussion is well-written.

We thank the referee for his very positive comment.

17. Studies of weight-reduced individuals suggest that following weight loss there is a significant improvement in muscle efficiency which would translate into an increased W/kg as described in this study. However, they also note a decline in resting energy metabolic rate/kg which should be detected in the SMR. The observation that the increase in relative SMR is not detected until post 2 is not consistent with the suggestion that there is a time in which muscle adaptation to walking (not standing) occurs during which the increased efficiency of muscle may be "masked" by lack of optimal walking style. Specifically, why should mechanical gait changes during the follow-up affect standing metabolic rate.

The non-consistent results in the relative SMR have now been better and deeply discussed in the manuscript. Please see pp. 23-24.

{The increase in the relative SMR ($W \cdot kg^{-1}$) at post 2 (Table 1), likely due to the substantial reduction in the metabolically inactive fat body mass, despite the loss of the total lean body mass, may partially contribute to the significant reduction in the mass-normalized NC_w only at post 2 (Browning et al., 2016). Although an increase in the relative SMR after a nonsurgical intervention (Peyrot et al., 2010) and in relative resting metabolic rate after bariatric surgery (de Cleva et al., 2018) has already been reported, the comparison of these mass-normalized measures of standing or resting metabolic rate (simple ratio-based assessments) before and after body mass loss is questioned (Cooper & Berman, 1994; Packard & Boardman, 1999; Browning et al., 2018). In fact, using body mass as covariate, we did not find any differences in the absolute SMR during follow-up, attesting that ratio might introduce problems with respect to statistical analysis and interpretation of SMR data (Browning et al., 2018). However, the results of the mixed linear model for the absolute NC_w with body mass as covariate confirmed those of the mass-normalized NC_w analysis (ratio analysis). This demonstrates that, for the main outcome of the study (the relative NC_w), i) the ratio analysis can be correctly used to analyse our data (Packard & Boardman, 1999) and ii) the decrease in body mass is not the only factor involved in the reduced relative NC_w after bariatric surgery. Another likely explanatory factor may be the mechanical gait changes during the follow-up.}

18. The implication is that SMR is representative of RMR raises a number of issues. RMR was used by Browning et al and the authors need to clarify the relationship between RMR and SMR in citing this study.

We thank the reviewer for the comment about the difference between SMR and RMR. According to this comment and for justifying this important methodological point, we added a paragraph in the Methods explaining why SMR is more relevant than RMR to assess the net energy cost of walking. Please see the paragraph “Energy cost of walking” p. 11.

{SMR is commonly used to calculate NC_w (Martin et al., 1992; Donelan et al., 2002; Browning et al., 2006; Browning & Kram, 2007; Massaad et al., 2007; Fernandez Menendez et al.,

2020) and, compared with basal metabolic rate, it better defines the metabolic cost associated with muscle contractions to maintain balance and support body weight while walking (Malatesta et al., 2003), which have to be considered for accurately assessing NC_w in individuals with normal weight (Malatesta et al., 2003) and with obesity (Peyrot et al., 2012).}.

19. The authors cite Vijgen et al, who noted improved mitochondrial function 1 year after bariatric surgery, but should also cite Fernstrom et al (Obes Surg, 2016), who made the same observation at 6 months.

The reference of Fernstrom et al. (Obes Surg, 2016) has now been added in the Discussion. Please see p. 27.

20. The authors conclude that a period of adaptation in walking style is necessary to see the improved efficiency of muscle. This is somewhat in contrast to Borges et al (Eur J Appl Physiol, 2018) who found a significant decrease in oxygen consumption/kg on the treadmill that was noted in premenopausal women who were not obese studied before and after weight loss. Also, data should be better contrasted with studies of dietary weight loss (e.g., the reference by Rosenbaum et al) which reported much earlier, but still significant, changes in muscle efficiency after a smaller degree of weight loss. This might reflect the different degrees of weight loss (and therefore greater degree of gait adjustment), different means of weight loss, etc.

We thank the reviewer for this important comment that allows us to modify our Discussion (pp. 27-28) and indirectly confirm that our very large body mass loss, inducing a greater degree of gait pattern change, would need a period of adaptation in the second part of the follow-up (from post 1 to post 2) to the new gait pattern, developed in the first part of the follow-up (from baseline to post 1), to improve its mechanical efficiency.

{However, some studies reported that the cycling efficiency (Rosenbaum et al., 2003) and NC_w (Borges et al., 2018) were improved directly after shorter body mass loss interventions and, thus, somewhat in contrast with our delayed enhancement in NC_w and mechanical

efficiency only at post 2. This might reflect the different degree of body mass loss obtained in individuals with overweight by these previous studies (-10-12 kg corresponding to 10-16% of the initial body mass) compared with that of the present study at post 1 (-30 kg or -25% of the body mass at baseline). Moreover, this difference indirectly confirms that a very large body mass loss, inducing a greater degree of gait pattern changes, would need a period of adaptation to the new gait pattern to improve its mechanical efficiency.}

Dear Dr Malatesta,

Re: JP-RP-2021-281710R1 "Effect of very large body mass loss on energetics, mechanics and efficiency of walking in adults with obesity: mass-driven vs behavioural adaptations" by Davide Malatesta, Julien Favre, Baptiste Ulrich, Didier Hans, Michel Suter, Lucie Favre, and Aitor Fernández Menéndez

Thank you for submitting your manuscript to The Journal of Physiology. It has been assessed by a Reviewing Editor and by 2 expert Referees and I am pleased to tell you that it is considered to be acceptable for publication following satisfactory revision.

The reports are copied at the end of this email. Please address all of the points and incorporate all requested revisions, or explain in your Response to Referees why a change has not been made.

NEW POLICY: In order to improve the transparency of its peer review process The Journal of Physiology publishes online as supporting information the peer review history of all articles accepted for publication. Readers will have access to decision letters, including all Editors' comments and referee reports, for each version of the manuscript and any author responses to peer review comments. Referees can decide whether or not they wish to be named on the peer review history document.

I hope you will find the comments helpful and have no difficulty returning your revisions within 4 weeks.

Your revised manuscript should be submitted online using the links in Author Tasks Link Not Available.

Any image files uploaded with the previous version are retained on the system. Please ensure you replace or remove all files that have been revised.

REVISION CHECKLIST:

- Article file, including any tables and figure legends, must be in an editable format (eg Word)
- Upload each figure as a separate high quality file
- Upload a full Response to Referees, including a response to any Senior and Reviewing Editor Comments;
- Upload a copy of the manuscript with the changes highlighted.

- A potential 'Cover Art' file for consideration as the Issue's cover image;
- Appropriate Supporting Information (Video, audio or data set <https://jp.msubmit.net/cgi->

bin/main.plex?form_type=display_requirements#supp).

To create your 'Response to Referees' copy all the reports, including any comments from the Senior and Reviewing Editors, into a Word, or similar, file and respond to each point in colour or CAPITALS and upload this when you submit your revision.

I look forward to receiving your revised submission.

If you have any queries please reply to this email and staff will be happy to assist.

Yours sincerely,

Michael C. Hogan
Senior Editor
The Journal of Physiology
<https://jp.msubmit.net>
<http://jp.physoc.org>
The Physiological Society
Hodgkin Huxley House
30 Farringdon Lane
London, EC1R 3AW
UK
<http://www.physoc.org>
<http://journals.physoc.org>

EDITOR COMMENTS

Reviewing Editor:

The reviewers' comments have been appropriately addressed and the manuscript has been accordingly improved. However, I invite the authors to consider the additional analysis suggested by Reviewer 1.

REFEREE COMMENTS

Referee #1:

The authors have addressed my concerns and I am largely satisfied with their responses. I thank them for addressing the issues raised.

I remain somewhat skeptical due to the potential for changes in SMR to affect calculated walking efficiency. I understand the authors' response and respect their reasoning, but it still seems possible that an artifact of increased SMR could cause a spurious decrease in calculated

walking cost at post-2.

The authors have revised the Discussion to address these possibilities. I would suggest one more analysis: does the change in walking economy at post-2 correlate with the change in walking economy? If so, that would suggest the possible impact of SMR artifact, and should be addressed in the Discussion. If not, it strengthens their conclusions.

On the other hand, I would note that the SMR/lean mass ratio is lowest at post-1 and returns to baseline levels at post-2. That could indicate that energy compensation in response to weight loss (reduced SMR - see work by Liebel and Rosenbaum, and more recently by Kevin Hall with the Biggest Loser studies) could be greater in post-1, and that this effect weakens as subjects bodies adjust to their new weight and composition. That would strengthen the argument that SMR does *not* have an artifact problem. Since the authors are addressing the potential for SMR artifact in the revise paper they might include this line of thinking in their revision as it strengthens their case.

Referee #2:

The authors have addressed previous comments.

END OF COMMENTS

1st Confidential Review

22-Jun-2021

EDITOR

Dear Dr. Hogan,

Thank you for your correspondence dated 12th of July 2021. We have addressed the criticisms and marked the responses and edits in blue in the reviewed version of the manuscript.

We thank the reviewer for his helpful comments, which improved the paper, and we hope that the manuscript is now suitable for publication in Journal of Physiology.

Do not hesitate to contact me for any further queries.

Sincerely,

Davide Malatesta on behalf of the co-authors

EDITOR COMMENTS

Reviewing Editor

The reviewers' comments have been appropriately addressed and the manuscript has been accordingly improved. However, I invite the authors to consider the additional analysis suggested by Reviewer 1.

We thank the editor for his positive comment. We considered and performed the additional analysis suggested by Referee #1 and modified the manuscript accordingly (please see the response to referee #1).

Referee #1

Thank you very much for reviewing our manuscript and for your valuable comments. We hope that the revisions will satisfy your standard.

We have provided our responses to your comments. Amended sentences are *in italic* with the additional wordings in blue.

[R1] = reviewer (comments).

[A] = authors (responses).

{...} = *text modified in the revised manuscript*.

Number of pages is referred to the corrected word version.

The authors have addressed my concerns and I am largely satisfied with their responses. I thank them for addressing the issues raised.

We thank the referee for his positive comment.

1. I remain somewhat skeptical due to the potential for changes in SMR to affect calculated walking efficiency. I understand the authors' response and respect their reasoning, but it still seems possible that an artifact of increased SMR could cause a spurious decrease in calculated walking cost at post-2. The authors have revised the Discussion to address these possibilities. I would suggest one more analysis: does the change in walking economy at post-2 correlate with the change in walking economy? If so, that would suggest the possible impact of SMR artifact, and should be addressed in the Discussion. If not, it strengthens their conclusions.

According to the referee's suggestion, we tested the correlation between the mass-normalized NCw and the change in the mass-normalized SMR between baseline and post 2 and we did not find a significant correlation. We have now reported this finding in the

Results (please see p. 20) and Discussion (please see p. 26) of the new version of the manuscript.

{There was no significant correlation between the mass-normalized NC_w (averaged values across all speeds) and the change in the mass-normalized SMR between baseline and post 2 ($r = 0.17$; $P = 0.66$).}

{However, there was no correlation between the difference in the mass-normalized SMR and the change in the relative NC_w ($J \cdot kg^{-1} \cdot m^{-1}$) between baseline and post 2.}

2. On the other hand, I would note that the SMR/lean mass ratio is lowest at post-1 and returns to baseline levels at post-2. That could indicate that energy compensation in response to weight loss (reduced SMR - see work by Liebel and Rosenbaum, and more recently by Kevin Hall with the Biggest Loser studies) could be greater in post-1, and that this effect weakens as subjects bodies adjust to their new weight and composition. That would strengthen the argument that SMR does *not* have an artifact problem. Since the authors are addressing the potential for SMR artifact in the revise paper they might include this line of thinking in their revision as it strengthens their case.

We thank the referee for this valuable comment and suggestion. We added this interesting point in the Discussion of the new version of the manuscript (please see p. 26).

{Moreover, compared to the baseline, the relative SMR per kg of lean body mass decreased only at post 1 and then returned to baseline levels at post 2 (Table 1). This could indicate that “metabolic adaptation” (Rosenbaum & Leibel, 2010) in response to weight loss could be greater in post 1, and that this effect weakened as participants’ bodies adjusted to their new body mass and composition, confirming previous findings after bariatric surgery (Knuth et al., 2014). This indirectly corroborates that the increase in the mass-normalized SMR at post 2 may not be the main determinant in the decrease in the mass-normalized NC_w at this time point.}

Dear Dr Malatesta,

Re: JP-RP-2021-281710R2 "Effect of very large body mass loss on energetics, mechanics and efficiency of walking in adults with obesity: mass-driven vs behavioural adaptations" by Davide Malatesta, Julien Favre, Baptiste Ulrich, Didier Hans, Michel Suter, Lucie Favre, and Aitor Fernández Menéndez

I am pleased to tell you that your paper has been accepted for publication in The Journal of Physiology.

NEW POLICY: In order to improve the transparency of its peer review process The Journal of Physiology publishes online as supporting information the peer review history of all articles accepted for publication. Readers will have access to decision letters, including all Editors' comments and referee reports, for each version of the manuscript and any author responses to peer review comments. Referees can decide whether or not they wish to be named on the peer review history document.

Are you on Twitter? Once your paper is online, why not share your achievement with your followers. Please tag The Journal (@jphysiol) in any tweets and we will share your accepted paper with our 23,000+ followers!

The last Word version of the paper submitted will be used by the Production Editors to prepare your proof. When this is ready you will receive an email containing a link to Wiley's Online Proofing System. The proof should be checked and corrected as quickly as possible.

Authors should note that it is too late at this point to offer corrections prior to proofing. The accepted version will be published online, ahead of the copy edited and typeset version being made available. Major corrections at proof stage, such as changes to figures, will be referred to the Reviewing Editor for approval before they can be incorporated. Only minor changes, such as to style and consistency, should be made a proof stage. Changes that need to be made after proof stage will usually require a formal correction notice.

All queries at proof stage should be sent to TJP@wiley.com

Yours sincerely,

Michael C. Hogan
Senior Editor
The Journal of Physiology
<https://jp.msubmit.net>
<http://jp.physoc.org>
The Physiological Society
Hodgkin Huxley House
30 Farringdon Lane
London, EC1R 3AW
UK
<http://www.physoc.org>
<http://journals.physoc.org>

P.S. - You can help your research get the attention it deserves! Check out Wiley's free Promotion Guide for best-practice recommendations for promoting your work at

www.wileyauthors.com/eeo/guide. And learn more about Wiley Editing Services which offers professional video, design, and writing services to create shareable video abstracts, infographics, conference posters, lay summaries, and research news stories for your research at www.wileyauthors.com/eeo/promotion.

* IMPORTANT NOTICE ABOUT OPEN ACCESS *

Information about Open Access policies can be found here
<https://physoc.onlinelibrary.wiley.com/hub/access-policies>

To assist authors whose funding agencies mandate public access to published research findings sooner than 12 months after publication The Journal of Physiology allows authors to pay an open access (OA) fee to have their papers made freely available immediately on publication.

You will receive an email from Wiley with details on how to register or log-in to Wiley Authors Services where you will be able to place an OnlineOpen order.

You can check if your funder or institution has a Wiley Open Access Account here
<https://authorservices.wiley.com/author-resources/Journal-Authors/licensing-and-open-access/open-access/author-compliance-tool.html>

Your article will be made Open Access upon publication, or as soon as payment is received.

If you wish to put your paper on an OA website such as PMC or UKPMC or your institutional repository within 12 months of publication you must pay the open access fee, which covers the cost of publication.

OnlineOpen articles are deposited in PubMed Central (PMC) and PMC mirror sites. Authors of OnlineOpen articles are permitted to post the final, published PDF of their article on a website, institutional repository, or other free public server, immediately on publication.

Note to NIH-funded authors: The Journal of Physiology is published on PMC 12 months after publication, NIH-funded authors DO NOT NEED to pay to publish and DO NOT NEED to post their accepted papers on PMC.

EDITOR COMMENTS

Reviewing Editor:

Thanks for these revisions that have improved the manuscript.

REFEREE COMMENTS

Referee #1:

I thank the authors for addressing my previous concerns and incorporating my suggestions into the revised text. I have no further comments.